# Infants are superior in implicit crossmodal learning and use other learning mechanisms than adults

**Sophie Rohlf[1][†]\*, Boukje Habets[1,2][†], Marco von Frieling[1], Brigitte Röder[1]**

[1]Biological Psychology and Neuropsychology, University of Hamburg, Hamburg, Germany; [2]Biological Psychology and Cognitive Neuroscience, University of Bielefeld, Bielefeld, Germany

**Abstract** During development internal models of the sensory world must be acquired which have to be continuously adapted later. We used event-related potentials (ERP) to test the hypothesis that infants extract crossmodal statistics implicitly while adults learn them when task relevant. Participants were passively exposed to frequent standard audio-visual combinations (A1V1, A2V2, p=0.35 each), rare recombinations of these standard stimuli (A1V2, A2V1, p=0.10 each), and a rare audio-visual deviant with infrequent auditory and visual elements (A3V3, p=0.10). While both six-month-old infants and adults differentiated between rare deviants and standards involving early neural processing stages only infants were sensitive to crossmodal statistics as indicated by a late ERP difference between standard and recombined stimuli. A second experiment revealed that adults differentiated recombined and standard combinations when crossmodal combinations were task relevant. These results demonstrate a heightened sensitivity for crossmodal statistics in infants and a change in learning mode from infancy to adulthood.
DOI: https://doi.org/10.7554/eLife.28166.001

**\*For correspondence:**
sophie.rohlf@uni-hamburg.de

[†]These authors contributed equally to this work

**Competing interests:** The authors declare that no competing interests exist.

## Introduction

After birth infants are immediately exposed to a sensory world comprising input of multiple sensory modalities. The developing brain must adapt to the statistical properties of the sensory environment (*Fiser et al., 2010*) since genetically defined neural circuits are usually crude. Indeed a high sensitivity of infants to statistical regularities within single sensory systems has often been demonstrated (*Fantz, 1964*; *Saffran et al., 1996*; *Fiser and Aslin, 2002*; *Bulf et al., 2011*). The seminal study of *Saffran et al. (1996)* reported that eight-month-old infants quickly learn transitional probabilities between syllables by pure exposure to an artificial language. This ability was interpreted as a basic mechanism allowing infants to segment a language. Similar results were found for non-linguistic auditory sequences and for visual patterns (*Saffran et al., 1999*; *Fiser and Aslin, 2002*), demonstrating a modality independent sensitivity of infants to statistical patterns in their sensory environment which moreover is not unique to linguistic material. For example, in the visual domain, there is strong evidence that infants are able to implicitly learn subtle statistical relationships among visual objects (*Fiser and Aslin, 2002*; *Bulf et al., 2011*; *Kirkham et al., 2002*). Nine-month-old infants who were exposed to multi element visual scenes, showed greater interest in element pairs which co-occurred more frequently than in pairs which co-occurred less frequently. Moreover, the infants were sensitive to the predictability between elements of the pairs as manifested by the conditional probability relations between these elements (*Fiser and Aslin, 2002*). The ability to extract statistical patterns of visual stimuli was found even in younger age groups (*Kirkham et al., 2002*); two-, five-, and eight-month-old infants were habituated to sequences of discrete visual stimuli whose ordering followed a statistical predictable pattern. Subsequently the infants were shown the previously encountered

**eLife digest** On a crowded city street, we automatically attribute the sounds of cars to the cars we see driving past, and not to the motorcycles or trucks on the same road. Similarly, we assign the voices we hear to the pedestrians around us, and not to the dogs those pedestrians are walking. As adults, we cope with these everyday challenges effortlessly, but how do infants first learn to match what they see with what they hear?

When young animals are exposed to new stimuli, their brains undergo changes. Similar changes only occur in adult animals if they deliberately pay attention to the stimuli and if they are associated with rewards. Rohlf, Habets et al. therefore predicted that human infants would automatically learn to associate sights and sounds upon being passively exposed to them. Adults, on the other hand, would learn these associations only if explicitly asked to do so.

To test this prediction, Rohlf, Habets et al. presented tones and colored shapes to 6-month-old infants and healthy adult volunteers while using scalp electrodes to monitor the electrical activity in their brains. Certain shapes and tones occurred frequently together, whereas other combinations of the same stimuli were rare. The 6-month-olds consistently outperformed the adults in associating the tones and shapes: the electrical activity in the infant brains reliably distinguished between common versus rare combinations. Adult brains made this distinction only when the adults were asked to pay attention to the tone-shape combinations as part of a task.

This high sensitivity to combinations of sights and sounds that regularly occur together enables infants to quickly learn about the world around them. As adults will have done this previously, the most effective strategy for adults is to update their existing knowledge only when such learning enables them to achieve a goal. Further research is needed to find out what happens in the brain to cause this change in learning strategy. Understanding how learning differs in infants and adults will help identify stages of development in which the brain learns particularly easily. This may ultimately help us optimize learning strategies for individuals of different ages.

DOI: https://doi.org/10.7554/eLife.28166.002

pattern alternating with a novel pattern of identical stimulus components. Infants of all age groups looked longer at the novel sequences providing evidence for the detection of visual statistical regularities at an early developmental stage. These results suggest that infants own powerful mechanisms for extracting the statistical properties of their sensory input without any instructions, explicit feedback, or intentional awareness (*Lany and Saffran, 2013*; *Krogh et al., 2012*).

The ability of infants to detect crossmodal statistical regularities within their sensory environment is less well understood, but some basic multisensory abilities, such as multisensory temporal synchrony detection seem to exist within the first month of life (*Lewkowicz, 1992*). In the next months the capability to perceive higher-level and more complex multisensory relations starts to develop. For example, at the age of six months infants were shown to perceive duration-based (*Lewkowicz, 1992*) and spatio-temporal based crossmodal relations (*Scheier et al., 2003*). Furthermore, there is evidence that similar to adults, infants take advantage of crossmodal events in terms of a better discrimination and a faster responsiveness to bimodal compared to unimodal information (*Bahrick et al., 2004*; *Lewkowicz and Kraebel, 2004*). First evidence for multisensory facilitation was found in eight-month-old infants as indicated by faster eye movements to spatially aligned auditory and visual cues compared to eye movements to each of these stimuli alone (*Neil et al., 2006*). Moreover, other studies revealed multisensory benefits for perceptual learning in infants (*Bahrick and Lickliter, 2000*; *Frank et al., 2009*). Five-month-old infants were habituated to either an audio-visual rhythm or the same rhythm presented unimodally. In the crossmodal condition, infants were able to discriminate between the familiar and a novel rhythm, whereas no discrimination was observed for the unimodal stimuli (*Bahrick and Lickliter, 2000*). Corresponding results were found for the learning of an abstract rule in five-month-old infants: they were able to learn the sequence if defined by redundant visual shapes and speech sounds but not if only one sensory modality was involved (*Frank et al., 2009*). These results suggest that infants are able to learn and use associations between auditory and visual stimuli. However, it must be taken into account that

the multisensory effects in infants were not tested against statistical facilitation (probability summation, see *Miller, 1982*).

Several studies on crossmodal association learning have reported that infants at the age of three months, but not younger, were able to learn specific voice-face combinations; infants were habituated to different unfamiliar voice-face pairings. In the post-familiarization test the infants showed higher attention to the learned voice-face pairs as compared to the novel combinations. The latter category comprised a voice and a face they had heard and seen previously, but the combination of the voice and face was new (*Brookes et al., 2001*; *Bahrick et al., 2005*). More recently, near-infrared spectroscopy (NIRS) and event-related potentials (ERPs) were used to test whether infants are able to learn crossmodal associations between arbitrary auditory and visual stimuli. *Emberson et al. (2015)* used an audio-visual omission paradigm with six-month-old infants and found similar visual cortex activation as a response to an auditory stimulus alone, which had been previously combined with a visual stimulus, as for the presentation of the same visual stimulus. The authors interpreted their findings as evidence for top-down mechanisms to be in place as early as six month of age. *Kouider et al. (2015)* exposed twelve-month-old infants to pictures of faces paired with one sound and pictures of flowers paired with a second sound. During the test phase the sound preceded the visual stimulus and was either congruent or incongruent with the learned combinations (additionally no sound was used in one third of the trials). An enhanced early positive ERP for congruent visual stimuli as well as an enhanced late negative ERP for incongruent visual stimuli were found. Both studies demonstrate that infants are able to learn crossmodal combinations to which they were exposed. However, none of these studies used an adult control group. Thus, it remains an open question of whether developmental and adult crossmodal learning recruit similar mechanisms. In this context it is interesting to notice that *Janacsek et al. (2012)* demonstrated superior implicit statistical learning of visual sequences in young children (<12 years) compared to older children and adults; a follow-up study indicated that this advantage was lost when they became more reliant on explicit learning (*Nemeth et al., 2013*).

Based on non-human animal studies it has been proposed (*Keuroghlian and Knudsen, 2007*) that developmental and adult plasticity, and thus learning, differ due to different brain states: during the sensitive phase molecular mechanisms dominate that allow for quick and extensive functional and structural synaptic plasticity (synaptogenesis, synaptic strengthening and elimination) allowing the emergence of a functional adaptive connectivity. By contrast, in adulthood these functionally tuned and to some degree stabilized neural circuits undergo adaptations when relevant to the system. Such age dependent changes from developmental to adult plasticity are impressively demonstrated by a study on auditory cortex plasticity in rats: while passive exposure to sounds of a specific frequency results in a permanent reorganization of auditory cortex during the sensitive phase, adult rats reorganized only those aspects of the auditory cortex which were task relevant: for example, rats were exposed to sounds which varied both in sound frequency and level. When they had to discriminate the sounds with respect to sound frequency the frequency representation of auditory cortex changed while the level representation changed when level rather than sound frequency was task relevant (*de Villers-Sidani et al., 2007*). These findings suggest that adult learning depend to a larger degree on attention and context such as task relevance and reward expectations (*Keuroghlian and Knudsen, 2007*; *Bavelier et al., 2010*). This hypothesis was supported by *Riedel and Burton (2006)* who investigated whether learning of auditory sequences is influenced by task demands; when using a serial reaction time task related to a feature of the auditory stimulus they found learning effects in adult participants while a passive exposure did not result in learning. Similarly, the statistical relations of concurrently presented visual streams were only learned by adults for the attended but not for the unattended streams (*Turk-Browne et al., 2005*). *Emberson et al. (2011)* extended these findings by providing evidence in adults that attention was necessary for implicit statistical learning in both the visual and auditory modality.

In the present study we investigated multisensory associative learning in infants and adults to test the hypothesis that infants as young as six months are not only able to learn arbitrary auditory-visual associations but that their sensitivity to crossmodal statistics is even higher compared to adults when crossmodal associations are passively encountered. Thus, in the first experiment we included a group of six-month-old infants (Experiment 1a) and a group of young adults (Experiment 1b). While recording the electroencephalogram (EEG), we presented two frequently occurring audio-visual standard combinations (A1V1, A2V2, p=0.35 each, 'Frequent standard stimuli'), two rare

recombinations of the 'Frequent standard stimuli' (A1V2, A2V1, p=0.10 each, 'Rare recombined stimuli') and one rare audio-visual combination of an infrequent auditory and an infrequent visual stimulus (A3V3, p=0.10, 'Rare deviant stimuli'). Recombining the auditory and visual elements of the 'Frequent standard stimuli' controls for the likelihood of the auditory and visual elements of the employed crossmodal stimuli. Thus, in order to detect 'Rare recombined stimuli' it is necessary to have learned the precise crossmodal combination. By contrast, the likelihood of both the visual and the auditory elements of 'Rare deviant stimuli' were lower than for all other auditory and visual elements. Therefore, the present experimental design allowed us to differentiate between the processing of the likelihood of sensory elements ('Frequent standard stimuli' vs. 'Rare deviant stimuli') and the processing of the conditional probabilities of crossmodal combinations ('Frequent standard stimuli' vs. 'Rare recombined stimuli').

We predicted ERP differences between the 'Frequent standard stimuli' and 'Rare deviant stimuli' in both infants (*Cheour et al., 2000*) und adults (*Schröger and Wolff, 1996*; *Näätänen and Alho, 1995*). In contrast, we hypothesized that only infants display an ERP difference for 'Frequent standard stimuli' vs. 'Rare recombined stimuli' due to a higher sensitivity to crossmodal statistics during infancy.

## Results

### Experiment 1

In Experiment 1 we investigated a group of infants (Experiment 1a) and a group of young adults (Experiment 1b) with the same experimental design. Due to the age difference between the groups, a few adjustments in the procedure, data recording, and data analyses were necessary.

### Experiment 1a (Infants)

ERP differences were found between 'Rare deviant stimuli' and 'Frequent standard stimuli' as well as 'Rare recombined stimuli' and 'Frequent standard stimuli': 'Rare deviant stimuli' (A3V3) elicited a more negative going ERP than 'Frequent standard stimuli' (A1V1, A2V2) (see *Figure 1*). This effect (200–420 ms, 420–1000 ms) was predominantly observed over the right hemisphere. Crucially, 'Rare recombined stimuli' (A1V2, A2V1) elicited a more negative going ERP compared to 'Frequent standard stimuli' (see *Figure 1*), predominantly over the left hemisphere (420–1000 ms).

#### First time window (200 – 420 ms): cluster analysis

The overall ANOVA with factors *Condition*, *Hemisphere*, and *Cluster* revealed a significant interaction between the factors *Condition* and *Hemisphere* ($F_{(2,56)}$ = 4.55; p=0.015) as well as a significant interaction of *Condition* × *Cluster* ($F_{(6,168)}$ = 4.94; p<0.001). Follow-up ANOVAs revealed a significant interaction of *Condition* × *Hemisphere* for cluster F ($F_{(2,56)}$ = 3.78; p=0.028), FC ($F_{(2,56)}$ = 3.67; p=0.029), and cluster CP ($F_{(2,56)}$ = 3.18; p=0.048). Post hoc t-tests showed that this interaction was driven by a more positive amplitude in response to 'Rare deviant stimuli' compared to 'Frequent standard stimuli' (see *Figure 1*) at cluster F ($t_{(28)}$ = 3.18; p=0.014), cluster FC ($t_{(28)}$ = 2.93; p=0.026), and cluster CP ($t_{(28)}$ = 3.02; p=0.02) of the right hemisphere.

#### First time window (200 – 420 ms): midline analysis

The overall ANOVA with factors *Condition* and *Electrode* showed a significant interaction between *Condition* x *Electrode* ($F_{(10,280)}$ = 2.76; p=0.002). Follow-up ANOVAs revealed a significant main effect of the factor *Condition* for electrode Fz ($F_{(2,56)}$ = 5.31; p=0.007) and FCz ($F_{(2,56)}$ = 3.79; p=0.02). Post hoc t-tests showed significant differences between the 'Rare deviant stimuli' and 'Frequent standard stimuli' at electrode FC ($t_{(28)}$ = 2.51; p=0.036) and FCz ($t_{(28)}$ = 2.45; p=0.04); 'Rare deviant stimuli' elicited a more positive going ERP than 'Frequent standard stimuli' (see *Figure 1*).

#### Second time window (420 – 1000 ms): cluster analysis

The overall ANOVA revealed a significant interaction of *Condition* × *Hemisphere* ($F_{(2,56)}$ = 4.68; p=0.013) as well as a significant interaction of *Condition* × *Cluster* ($F_{(6,168)}$ = 4.51; p<0.01). Follow-up ANOVAs separately calculated for each cluster showed a significant interaction of *Condition* × *Hemisphere* at Cluster F ($F_{(2,56)}$ = 4.5; p=0.014) and cluster FC ($F_{(2,56)}$ = 4.6; p=0.013). Post-hoc

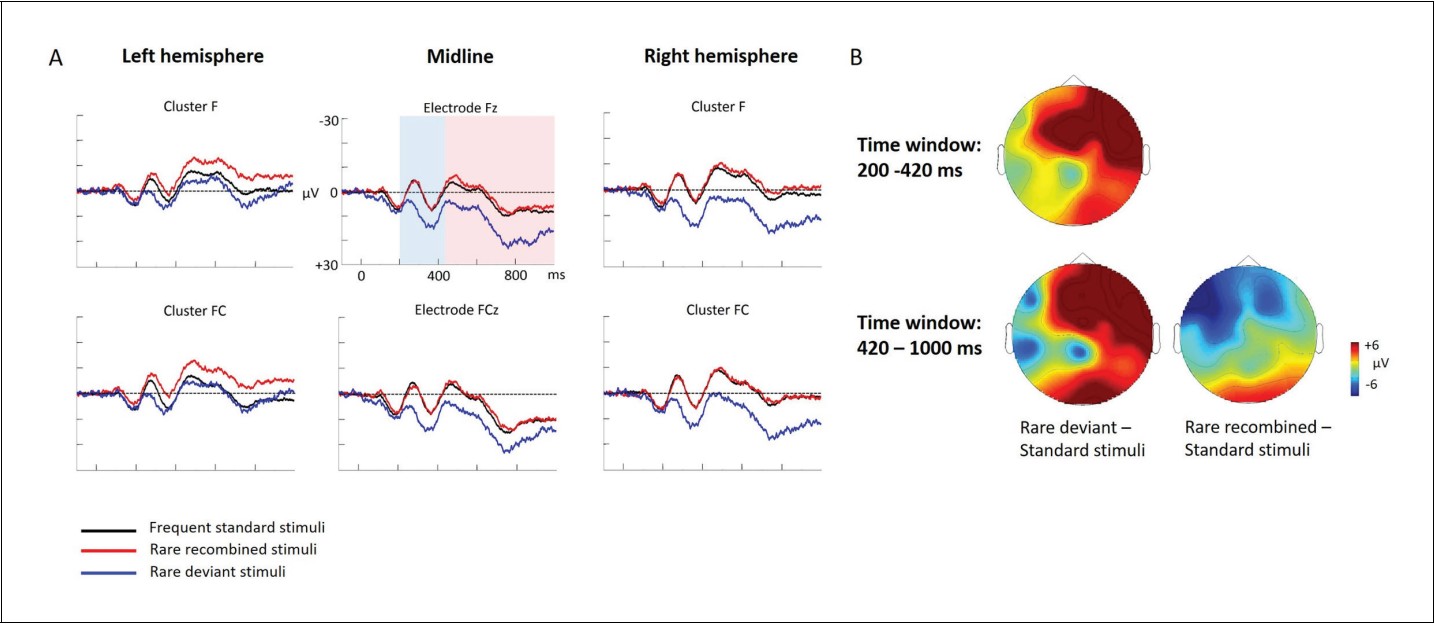

**Figure 1.** Grand average ERPs of Experiment 1a. (**A**) ERPs to the three conditions ('Frequent standard stimuli', 'Rare recombined stimuli', 'Rare deviant stimuli') are superimposed for the electrode clusters F and FC, and the electrodes Fz and FCZ. The analyzed time epochs are marked in blue (200–420 ms) and red (420–1000 ms). (**B**) The topographical distribution of the difference between 'Rare deviant stimuli' minus 'Frequent standard stimuli' and 'Rare recombined stimuli' minus 'Frequent standard stimuli' for the first and second time window.
DOI: https://doi.org/10.7554/eLife.28166.003

t-tests indicated that ERPs to 'Rare deviant stimuli' were significantly more positive than ERPs to 'Frequent standard stimuli' (see *Figure 1*) at cluster F (t(28) = 2.72; p=0.044) of the right hemisphere. In addition, post hoc t-tests revealed significant differences between 'Frequent standard stimuli' and 'Rare recombined stimuli' at cluster FC of the left hemisphere (t(28) = −2.81; p=0.032), indicating a more negative amplitude in response to 'Rare recombined stimuli' compared to the 'Frequent standard stimuli' (see *Figure 1*).

## Second time window (420 – 1000 ms): midline analysis
The ANOVA revealed a significant interaction between the factors *Condition* and *Electrode* (F (10,280) = 2.76; p=0.002). Follow-up ANOVAs indicated a main effect of *Condition* for electrode AFz (F(2,56) = 3.4; p=0.04) and Fz (F(2,56)= 3.59; p=0.03). However, none of the subsequent t-tests reached significance (all p≥0.08).

## Experiment 1b (Adults)
ERP differences were found only between 'Rare deviant stimuli' and 'Frequent standard stimuli'. ERPs to 'Rare deviant stimuli' were more negative going than ERPs to 'Frequent standard stimuli' during both time windows (180–220 ms, 250–1000 ms; see *Figure 2*).

## First time window (180 – 220 ms): cluster analysis
The overall ANOVA did not reveal any significant effect involving the factor *Condition.*

## First time window (180 – 220 ms): midline analysis
The overall ANOVA revealed a significant interaction between the factors *Condition* and *Electrode* (F(12,276) = 2.16; p=0.03). Follow-up ANOVAs obtained a significant main effect of *Condition* for electrode Cz (F(2,46) = 4.02; p=0.024). Post hoc t-tests showed significant differences between the 'Rare deviant stimuli' and 'Frequent standard stimuli' at electrode Cz (t(22) = −2.32; p=0.047); 'Rare deviant stimuli' elicited a more negative going ERP than 'Frequent standard stimuli' (see *Figure 2*).

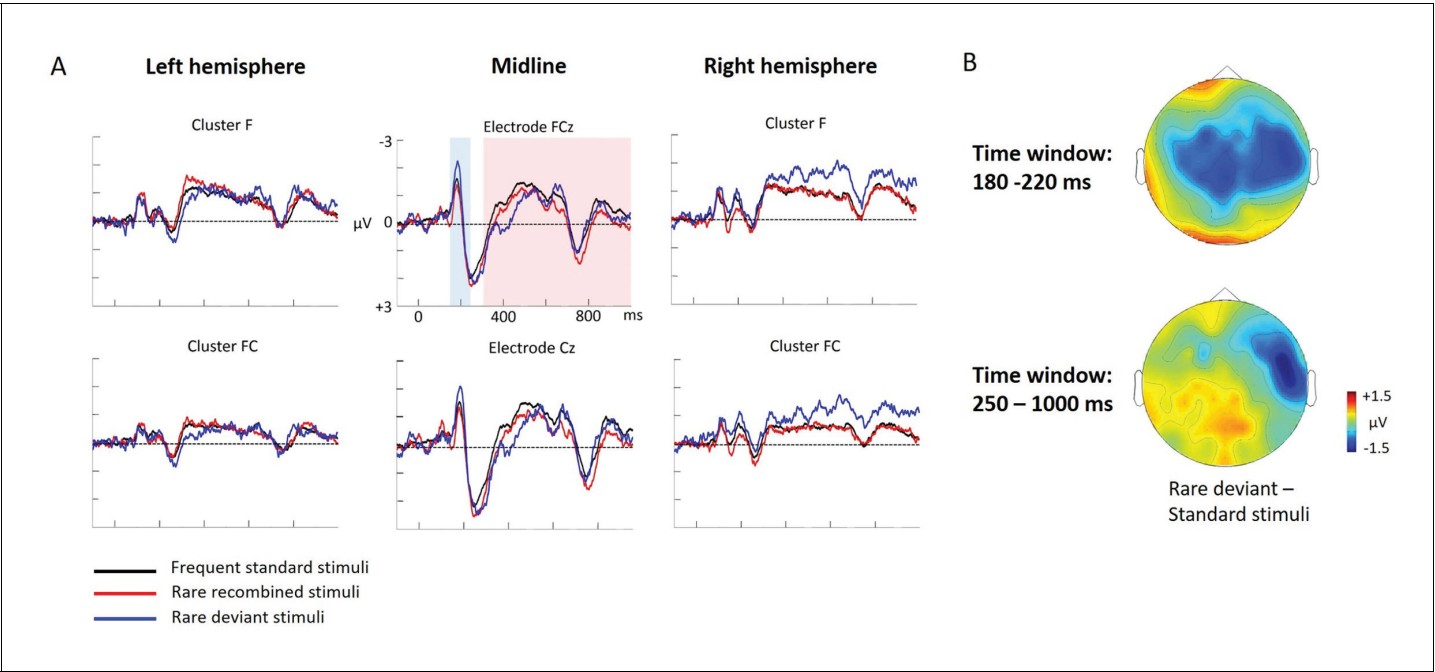

**Figure 2.** Grand average ERPs of Experiment 1b. (**A**) ERPs to the three conditions ('Frequent standard stimuli', 'Rare recombined stimuli', 'Rare deviant stimuli') are superimposed for the electrode clusters F and FC, and the electrodes FCz and Cz. The analyzed time epochs are marked in blue (180–220 ms) and red (420–1000 ms). (**B**) The topographical distribution of the difference between 'Rare deviant stimuli' minus 'Frequent standard stimuli' for the first and second time window.

DOI: https://doi.org/10.7554/eLife.28166.004

### Second time window (250 – 1000 ms): cluster analysis

The overall ANOVA revealed a significant interaction between the factors *Condition, Hemisphere,* and *Cluster* (F(10,230) = 2.49; p=0.007). Follow-up ANOVAs separately calculated for each cluster obtained a significant interaction of *Condition* and *Hemisphere* for cluster FC (F(2,46) = 4.56; p=0.015). Post hoc t-tests showed that this interaction was driven by a more negative amplitude in response to 'Rare deviant stimuli' compared to 'Frequent standard stimuli' (see *Figure 2*) at cluster FC of the right hemisphere (t(22) = −2.22; p=0.036).

### Second time window (250 – 1000 ms): midline electrodes

The overall ANOVA did not reveal any significant effect involving factor *Condition*.

## Summary and discussion of Experiment 1a and 1b

As predicted, infants were more sensitive to crossmodal statistics than adults. Only infants displayed a significant ERP deviant effect for 'Rare recombined stimuli'. By contrast, both groups showed at a relatively earlier time epoch a 'Rare deviant stimuli' effect, suggesting that the overall experimental power had been sufficient to detect ERP deviant effects. In fact, the effect size for the 'Rare deviant stimuli' effects was smaller, both in infants (*d* = 0.65) and adults (*d* = 0.63), than the effect size for the ERP effects comparing 'Frequent standard stimuli' and 'Rare recombined stimuli' in infants (*d* = 0.73). Thus, since smaller effects ('Frequent standard stimuli' vs. 'Rare deviant stimuli') than the missing effect ('Frequent standard stimuli' vs. 'Rare recombined stimuli') were detected in adults, it seems justified to conclude that the null effect in adults was not caused by a lack of power. Nevertheless, we ran a second Experiment (Experiment 2a) to replicate with a more adequate design for adults the lack of learning arbitrary crossmodal conditional probabilities when they were not related to a task. Moreover, in an additional experiment (Experiment 2b) we tested the requirements for adult learning of crossmodal statistics. We will discuss the results of Experiment 1a and 1b in light of the results of these additional experiments in the general Discussion.

## Experiment 2

As we did not find any ERP difference between 'Frequent standard stimuli' and 'Rare recombined stimuli' in the adult group in Experiment 1b, we ran a second study in adults comprising two experiments, in which we systematically manipulated the task relevance of crossmodal combinations. Both experiments were very similar to Experiment 1 but comprised essential adaptations: (a) to enhance the power of the experiment, we increased the number of trials; (b) in Experiment 2a we included a fourth visual stimulus (V4), which had to be detected by participants (target) while all other stimuli remained task irrelevant: This manipulation guaranteed that participants attended the stimuli; (c) in Experiment 2b one of the 'Rare recombined stimuli' (either A1V2 or A2V1) served as the target: this manipulation rendered crossmodal combinations task relevant to the participants. At the same time this design allowed us to analyze ERPs to crossmodal stimuli, including to the non-target 'Rare recombined stimuli', which were, as in Experiment 2a, not followed by a manual response.

We hypothesized that adults are not sensitive to crossmodal statistics (no ERP difference between 'Frequent standard stimuli' and 'Rare recombined stimuli') when crossmodal combinations are task irrelevant (Experiment 2a in replication of the findings from Experiment 1b) but that such ERP differences would emerge in Experiment 2b, indicating learning of crossmodal statistics when they are task relevant.

## Behavioral data

As seen in *Table 1*, participants identified target stimuli with a high accuracy in both experiments.

## Experiment 2a: ERP data

ERP differences were found only between 'Rare deviant stimuli' and 'Frequent standard stimuli': Compared to 'Frequent standard stimuli' 'Rare deviant stimuli 'elicited a more negative early ERP (80–160 ms, see *Figure 3*). During the late time window (250–850 ms) ERPs to 'Rare deviant stimuli 'were more negative over the anterior scalp and more positive over the posterior scalp compared to ERPs to 'Frequent standard stimuli'. ERPs to 'Frequent standard stimuli' and 'Rare recombined stimuli' did not significantly differ (see *Figure 3*).

### First time window (80 – 160 ms): cluster analysis

The overall ANOVA revealed a significant interaction between the factors *Condition* and *Cluster* ($F_{(10,110)} = 4.71$; $p<0.001$). Follow-up ANOVAS separately calculated for each Cluster showed a significant main effect of *Condition* for cluster C ($F_{(2,22)} = 29.52$; $p<0.001$). Post-hoc t-tests indicated that ERPs to 'Rare deviant stimuli' were significantly more negative than ERPs to 'Frequent standard stimuli' (see *Figure 3*) at cluster C ($t_{(11)} = -5.76$; $p<0.001$).

### First time window (80 – 160 ms): midline analysis

The overall ANOVA revealed a significant interaction of *Condition* × *Electrode* ($F_{(12,132)} = 7.03$; $p<0.001$). Follow-up ANOVAs for each electrode obtained a significant main effect of *Condition* for electrode FCz ($F_{(2,22)} = 21.97$; $p<0.001$), Cz ($F_{(2,22)} = 21.74$; $p<0.001$), CPz ($F_{(2,22)} = 26.36$; $p<0.001$) and Pz ($F_{(2,22)} = 16.92$; $p<0.001$). Subsequent t-tests showed that this main effect was driven by a significant more negative amplitude in response to the 'Rare deviant stimuli' compared to the 'Frequent standard stimuli' (see *Figure 3*) at electrode FCz ($t_{(11)} = -5.82$; $p=0.003$), Cz ($t_{(11)} = -5.39$; $p=0.001$), CPz ($t_{(11)} = -5.41$; $p<0.001$), and Pz ($t_{(11)} = -3.62$; $p=0.006$).

**Table 1.** Mean (±SEM) of reaction times (in ms), hit rates (in %), misses (in %), and false alarms (in %) to the target stimuli of Experiment 2a and Experiment 2b.

|  | RT (ms) | Hits (%) | Misses (%) | False alarms (%) |
| --- | --- | --- | --- | --- |
| Experiment 2a | 391 ± 17.5 | 99.4 ± 0.3 | 0.34 ± 0.18 | 0.63 ± 0.25 |
| Experiment 2b | 535 ± 27.5 | 96.6 ± 1.6 | 3.4 ± 1.6 | 15.55 ± 6.95 |

DOI: https://doi.org/10.7554/eLife.28166.005

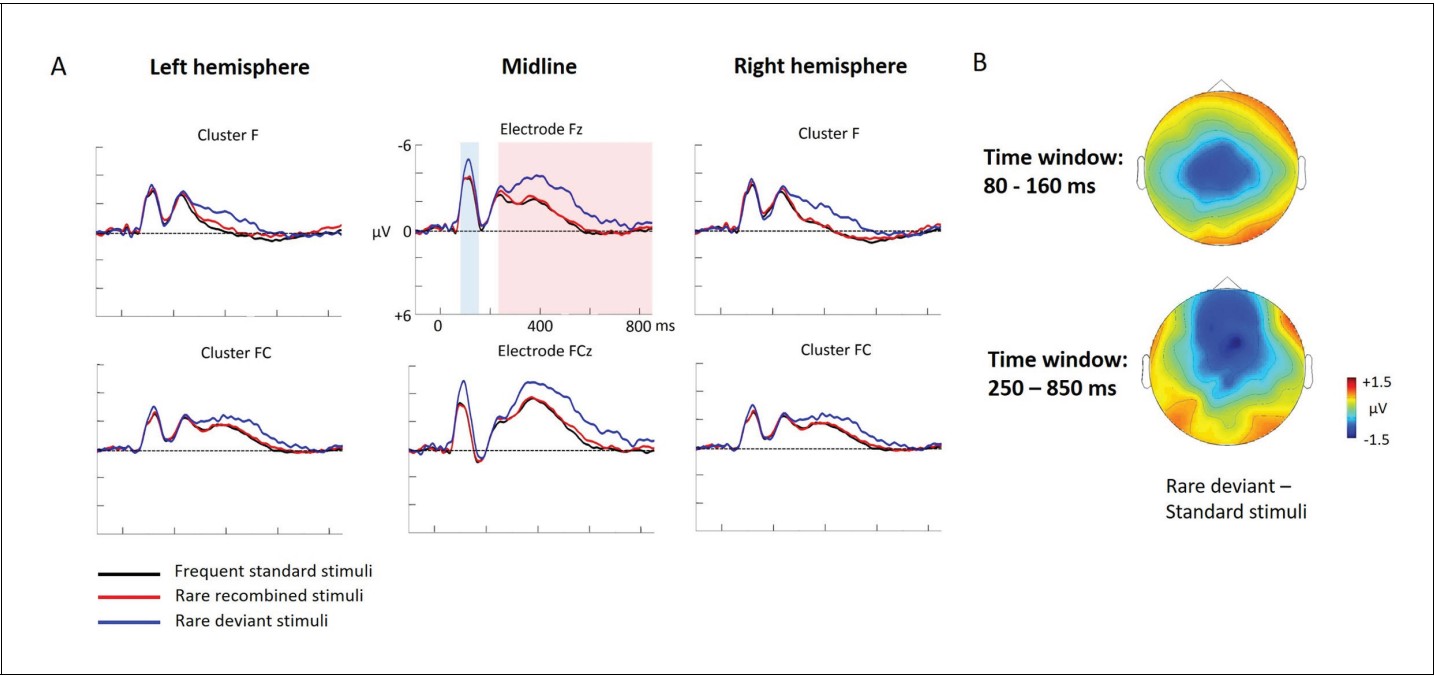

**Figure 3.** Grand average ERPs of Experiment 2a. (A) ERPs to the three conditions ('Frequent standard stimuli', 'Rare recombined stimuli', 'Rare deviant stimuli') are superimposed for the electrode clusters F and FC, and the electrodes Fz and FCZ. The analyzed time epochs are marked in blue (80–160 ms) and red (250–850 ms). (B) The topographical distribution of the difference between 'Rare deviant stimuli' minus 'Frequent standard stimuli' for the first and second time window.

DOI: https://doi.org/10.7554/eLife.28166.006

### Second time window (250 – 850 ms): cluster analysis

The overall ANOVA revealed an interaction between *Condition* and *Cluster* ($F_{(10,110)}$ = 3.23; p<0.001). Follow-up ANOVAs separately calculated for each cluster showed a significant main effect of factor *Condition* for cluster P ($F_{(2,22)}$ = 4.9; p=0.015) and cluster PO ($F_{(2,22)}$ = 4.74; p=0.017). Post-hoc t-tests indicated that ERPs in response to 'Rare deviant stimuli' were significantly more positive compared to ERPs to 'Frequent standard stimuli' (see *Figure 3*) at cluster P ($t_{(11)}$ = 3.46; p=0.008) and cluster PO ($t_{(11)}$ = 3.47; p=0.008)

### Second time window (250 – 850 ms): midline analysis

The overall ANOVA revealed a significant interaction of *Condition* × *Electrode* ($F_{(12,132)}$ = 3.82; p<0.001). Sub ANOVAs for each electrode showed a significant main effect for the factor *Condition* at electrode Fz $_{(2,22)}$=10.59; p<0.001), FCz ($F_{(2,22)}$= 8.86; p=0.001), Cz ($F_{(2,22)}$ = 4.13; p=0.027). Subsequent t-tests detected significant differences between the 'Frequent standard stimuli' and 'Rare deviant stimuli' at electrode Fz ($t_{(11)}$ = −5.71; p<0.001), FCz ($t_{(11)}$ = −4.49; p=0.001), and Cz ($t_{(11)}$ = −2.53; p=0.049); ERPs to 'Rare deviant stimuli' were more negative going than ERPs to 'Frequent standard stimuli' (see *Figure 3*).

## Experiment 2b: ERP data

ERP differences were found between both, 'Rare deviant stimuli' and 'Frequent standard stimuli' and between 'Rare recombined stimuli' and 'Frequent standard stimuli'. Compared to 'Frequent standard stimuli' 'Rare deviant stimuli 'elicited a more negative early ERP (80–160 ms, see *Figure 4*) over the anterior scalp and a more positive ERP over the posterior scalp. During the late time window (250–850 ms, see *Figure 4*) ERPs to 'Rare deviant stimuli 'were more positive over the anterior scalp and more negative over the posterior scalp compared to the 'Frequent standard stimuli'. ERPs to 'Rare recombined stimuli' compared to ERPs to 'Frequent standard stimuli' were more positive going over the anterior scalp and more negative going over the posterior scalp (250–850 ms, see *Figure 4*).

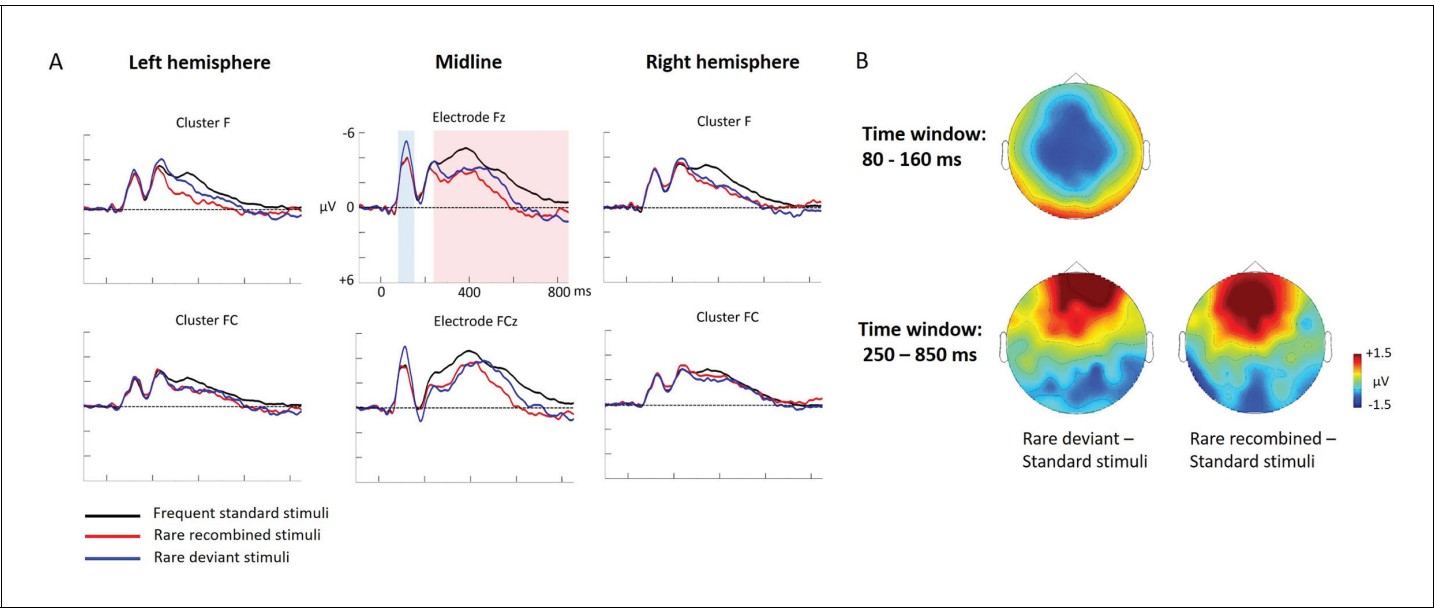

**Figure 4.** Grand average ERPs of Experiment 2b. (**A**) ERPs to the three conditions ('Frequent standard stimuli', 'Rare recombined stimuli', 'Rare deviant stimuli') are superimposed for the electrode clusters F and FC, and the electrodes Fz and FCZ. The analyzed time epochs are marked in blue (80–160 ms) and red (250–850 ms). (**B**) The topographical distribution of the difference 'Rare deviant stimuli' minus 'Frequent standard stimuli' and 'Rare recombined stimuli' minus 'Frequent standard stimuli' for the first and second time window.
DOI: https://doi.org/10.7554/eLife.28166.007

### First time window (80 – 160 ms): cluster analysis

The overall ANOVA revealed a significant interaction of *Condition* × *Cluster* ($F_{(10,110)}$ = 3.82; p=0.044). Further sub-ANOVAs separately calculated for each cluster showed a main effect of *Condition* for cluster C ($F_{(2,22)}$ = 5.83; p=0.003) and cluster PO ($F_{(2,22)}$ = 4.16; p=0.027), indicating a significant more negative amplitude in response to 'Rare deviant stimuli' than to ''Frequent standard stimuli' (see *Figure 4*) at cluster C ($t_{(11)}$ = −4.44; p=0.001) and a more positive amplitude in response to 'Rare deviant stimuli' compared to 'Frequent standard stimuli' at cluster PO ($t_{(11)}$ = 3.19; p=0.014).

### First time window (80 – 160 ms): midline analysis

The overall ANOVA revealed a significant interaction of *Condition* × *Electrode* ($F_{(12,132)}$ = 2.72; p=0.002). Follow-up ANOVAs for each electrode revealed a significant main effect of *Condition* for electrode FCz ($F_{(2,22)}$ = 4.28; p=0.024), Cz ($F_{(2,22)}$ = 6.01; p=0.007) and CPz ($F_{(2,22)}$ = 3.67; p=0.039). Subsequent t-tests indicated that ERPs to 'Rare deviant stimuli' were more negative than to 'Frequent standard stimuli' (see *Figure 4*) at electrode FCz ($t_{(11)}$ = −2.85; p=0.026), Cz ($t_{(11)}$ = −3.59; p=0.006), and CPz ($t_{(11)}$ = −2.59; p=0.044).

### Second time window (250 – 850 ms): cluster analysis

The overall ANOVA revealed a significant interaction of *Condition* × *Cluster* ($F_{(10,110)}$ = 4.12; p<0.001). Follow-up ANOVAs separately calculated for each Cluster showed a significant main effect of *Condition* for cluster F ($F_{(2,22)}$=5.09; p=0.013), FC ($F_{(2,22)}$ = 4.4; p=0.022), CP ($F_{(2,22)}$ = 6.42; p=0.005), and PO ($F_{(2,22)}$ = 6.35; p=0.005). Subsequent t-tests indicated significant more positive going ERPs to 'Rare deviant stimuli' than to 'Frequent standard stimuli' (see *Figure 4*) at cluster F ($t_{(11)}$ = 2.77; p=0.03) and FC ($t_{(11)}$ = 3.88; p=0.004), CP ($t_{(11)}$ = 2.62; p=0.041) and more negative going ERPs PO ($t_{(11)}$ = −3.61; p=0.01). In addition, t-tests showed that ERPs to 'Rare recombined stimuli' were more positive going than to 'Frequent standard stimuli' (see *Figure 4*) at cluster F ($t_{(11)}$ = 3.11; p=0.016), CP ($t_{(11)}$ = 3.43; p=0.009), and more negative going at PO ($t_{(11)}$ = −3.41; p=0.016).

## Second time window (250 –850 ms): midline analysis

The overall ANOVA revealed a significant interaction between *Condition and Electrode* ($F(12,132)$ = 7.62; p<0.001). Follow-up ANOVAs separately calculated for each electrode showed a main effect of *Condition* for electrode Fz ($F(2,22)$ = 7.42; p=0.003), FCz ($F(2,22)$ = 9.24; p<0.001), Cz ($F(2,22)$ = 9.24; p<0.001), Pz ($F(2,22)$ = 6.49; p=0.005), POz ($F(2,22)$ = 7.92; p=0.002), and Oz ($F(2,22)$ = 5.62; p=0.009). Subsequent t-tests indicated that ERPs to 'Rare deviant stimuli' were more positive going than to 'Frequent standard stimuli' (see *Figure 4*) at electrode Fz ($t(11)$ = 2.86; p=0.013), FCz ($t(11)$ = 3.71; p=0.002) and more negative at Pz ($t(11)$ = −3.23; p=0.006), POz ($t(11)$ = −2.93; p=0.01), and Oz ($t(11)$ = −2.54; p=0.024). Additionally, t-tests confirmed more positive going ERPs to 'Rare recombined stimuli' than to 'Frequent standard stimuli' (see *Figure 4*) at electrode Fz ($t(11)$ = 3.54; p=0.01) and FCz ($t(11)$ = 4.29; p=0.002) and more negative going ERPs at electrodes Pz ($t(11)$ = −3.49; p=0.003), POz ($t(11)$ = −3.58; p=0.006), and Oz ($t(11)$ = −3.29; p=0.01).

## Summary and discussion of Experiment 2a and 2b

Differences in ERPs between 'Rare deviant stimuli' and 'Frequent standard stimuli' were found in both experiments at early processing stages. Crucially, ERP differences between 'Rare recombined stimuli' and 'Frequent standard stimuli' were only found in Experiment 2b, indicating that the adults' brains were able to differentiate 'Rare recombined stimuli' from 'Frequent standard stimuli' when crossmodal combinations were task relevant.

## Discussion

The goal of the present study was to test for a higher sensitivity of infants as compared to adults to crossmodal statistics and to compare the mechanisms of crossmodal association learning in infants and adults. We conducted four ERP experiments in which infants and adults were exposed to audio-visual stimulus combinations with different probabilities. We presented 'Frequent standard stimuli' (A1V1, A2V2, p=0.35 each), rare recombinations of the 'Frequent standard stimuli' (A1V2, A2V1, p=0.10 each, 'Rare recombined stimuli'), and a rare deviant audio-visual combination with an infrequent auditory and visual element (A3V3, p=0.10, 'Rare deviant stimuli'). While infants passively learned the crossmodal combinations, adults did not. Adults' ERPs to 'Rare recombined stimuli' and to 'Frequent recombined stimuli' differed only when crossmodal combinations were task relevant. In contrast, all groups, irrespectively of learning context, showed a sensitivity to the probability of sensory elements, that is, for 'Rare deviant stimuli'. *Table 2* graphically summarizes the main results of all four experiments.

Studies using artificial languages or visual artificial scenes have repeatedly demonstrated that infants develop a sensitivity to the likelihood of events as well as to conditional probabilities (*Krogh et al., 2012*; *Aslin, 2014*), partially as early as at the age of two months (*Kirkham et al., 2002*). Two recent studies addressing crossmodal statistical learning found that six-month and twelve-month-old infants learned to predict a visual stimulus based on a preceding auditory stimulus (*Emberson et al., 2015*; *Kouider et al., 2015*). While *Kouider et al. (2015)* demonstrated that infants at the age of twelve months were able to learn an association between an arbitrary sound and a visual object category (faces vs. flowers), they did not include an adult control group and were thus not able to demonstrate differences in learning between adults and infants, nor were they able to distinguish processes related to the detection of crossmodal combinations and processes related to the familiarity with certain sensory elements.

Thus, the present study extends previous research by showing that the conditional probabilities of crossmodal combinations were extracted by infants as young as six months after a short exposure period while adults failed to learn crossmodal statistics under this condition. It is important to notice that we controlled for the likelihood of the auditory and visual elements of the employed crossmodal stimuli by infrequently recombining the auditory and visual elements of the 'Frequent standard stimuli'. We provide ERP evidence demonstrating that the processing of the conditional probabilities of crossmodal combinations and the processing of the likelihood of sensory elements can be dissociated: in infants, 'Rare recombined stimuli' elicited a left negative potential starting at about 420 ms post-stimulus while 'Rare deviant stimuli' elicited a right lateralized positivity starting at 200 ms post-stimulus (Experiment 1a). Adults tested under identical conditions were only sensitive to 'Rare deviant stimuli', which differed from 'Frequent standard stimuli' in the frequency of their auditory and

**Table 2.** Summary of the main results and topographical distributions of the two effects of interest.

(a) 'Rare deviant stimuli' minus 'Frequent standard stimuli' and (b) 'Rare recombined stimuli' minus 'Frequent Standard stimuli') in Experiment 1a, 1b, 2a and 2b. Electrodes and electrode clusters with significant differences between the experimental conditions are marked with black asterisks, comparisons with no significant differences are indicated by n.s..

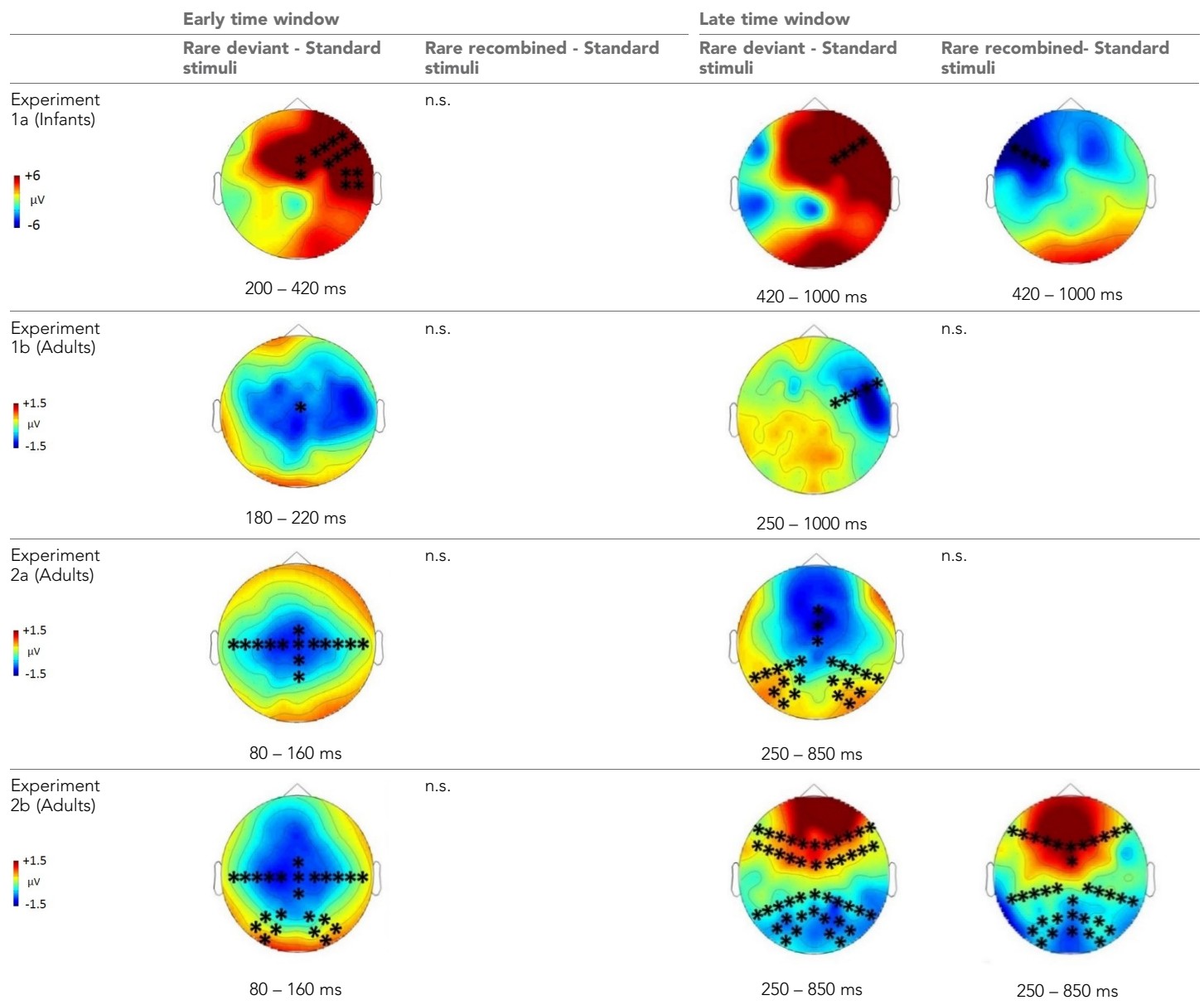

DOI: https://doi.org/10.7554/eLife.28166.008

visual elements (Experiment 1b, ERP effect starting 180 ms post-stimulus) but not for rare crossmodal stimuli which only differed from the 'Frequent standard stimuli' in the way the auditory and visual elements were combined. These results demonstrate that infants were able to learn arbitrary crossmodal associations as early as six months of age and thus much earlier than suggested by the study of *Kouider et al. (2015)* (see *Emberson et al., 2015*). This finding is in line with behavioral studies employing natural stimuli, which showed that infants from three months onwards were able to learn specific face-voice-parings (*Brookes et al., 2001*; *Bahrick et al., 2005*). Our results extended these behavioral findings by providing first evidence that the learning of crossmodal statistics in infancy is particularly sensitive and superior to adults when crossmodal stimuli are not task relevant. It could be argued that the signal to noise ratio of the ERPs in adults was not sufficient in Experiment 1b to

demonstrate crossmodal learning in the adult group. However, such an account is highly unlikely given that an effect of smaller size were detected in Experiment 1b and the fact, that in Experiment 2a an ERP difference between 'Frequent standard stimuli' and 'Rare recombined stimuli' was not significant either despite a much higher signal to noise ratio in comparison to Experiment 1b.

Thus, our results provide evidence that crossmodal statistics are better implicitly learned in the developing than in the adult system. An enhanced sensitivity for low-level statistical patterns during development compared to adulthood has been reported by other studies as well. For example, *Janacsek et al. (2012)* and *Nemeth et al. (2013)* demonstrated that children are superior in implicit statistical learning of sequences compared to adults but later lose this advantage and become more reliant on explicit learning. A similar developmental time course was found in a study of *Jost et al., 2011*, who had investigated the neurophysiological correlates of visual statistical learning in children and adults: children showed learning related ERP effects earlier in the acquisition phase than the adult group indicating that they had quicker acquired the statistical structurer. It is, however, important to notice that not all studies investigating statistical learning during development have found enhanced learning performance in infants or children as compared to adults. For example, *Saffran et al. (1996)*, *1999*) reported similar abilities in eight-month-old infants and adults in the extraction of the underlying statistical structure of auditory sequences. Other studies observed better learning for older children and young adults than in younger age groups (*Maybery et al., 1995*; *Fletcher et al., 2000*; *Kirkham et al., 2007*). *Arciuli and Simpson (2011)* tested children between the age of 5 and 12 years in a visual triplet learning task and reported better learning with increasing age. At first glance, these findings seem to be at odds with the present results. However, a closer look at the employed paradigms suggests that these different outcomes might be related to the complexity of the implemented statistical patterns. For example, *Arciuli and Simpson (2011)* tested the incidental learning of four visual base triples which draws to a much larger extend from working memory than learning the conditional probability of two audio-visual combinations as used in the present and previous crossmodal learning studies in infants (*Emberson et al., 2015*; *Kouider et al., 2015*). Indeed, it is well known that working memory improves during childhood (*Zelazo et al., 2008*) and thus working memory demands might have been the limiting factor in some studies (e.g. *Arciuli and Simpson, 2011*). In addition, triples are usually embedded in a continuous stream while the crossmodal stimuli of the present study were individually presented with relatively long interstimulus intervals, thereby clearly demarking the individual events. Furthermore, studies have revealed that the ability to extract statistical patterns from sensory input during infancy improves from the simple tracking of event probabilities early in the development (from three months onwards, see *Fantz, 1964*) to the learning of more complex and higher-level statistical patterns at a later developmental stage (from twelve months onwards, see *Gómez and Maye, 2005*). Thus, what most likely declines during development seems to be the sensitivity to simple conditional probabilities (*Janacsek et al., 2012*). *Janacsek et al. (2012)* speculated that a decline in the sensitivity to 'base probabilities' is necessary for the acquisition of higher order representations and a switch to model-based (in contrast to model-free) learning.

In line with this suggestion, adults did not learn crossmodal statistics when they were irrelevant for the task but became sensitive to them when a specific crossmodal combination was of behavioral relevance. Studies in non-human animals have suggested that during the sensitive phase, neural networks are elaborated in response to a pure exposure to the environment while during later development and in adulthood learning is context-specific and depends on task relevance (e.g. reward) and instructions (*Keuroghlian and Knudsen, 2007*). Currently, we can only speculate about the neural underpinnings of the age-dependent neuroplasticity as observed in the present study. As noted by *Dehaene-Lambertz and Spelke (2015)* feedforward connectivity seems to be to a larger degree genetically determined than feedback connectivity and the latter seems to be mostly experience dependent. The detection of 'Rare recombined stimuli' was associated with a relatively late ERP effect in both infants and adults. Indeed, multisensory binding has been found to rely on later processing stages in adults and the involvement of feedback connections (*Bruns and Röder, 2010*; *Bonath et al., 2007*). *Emberson et al. (2015)* provided evidence that the crossmodal connectivity is at least partially in place at the age of six months. Here we speculate that this initial crossmodal connectivity might even be more extensive in the developing brain (see *Johannsen and Röder, 2014*) and thus might be the neural underpinning of the enhanced sensitivity to simple crossmodal statistics in development which allows for quicker and a passive learning during infancy. We further

assume in line with the 'multisensory perceptual narrowing' idea (*Lewkowicz and Ghazanfar, 2006*) that experience narrows down the initial crossmodal connectivity by eliminating non-confirmed connections while elaborating connections which are useful for an individual (*Johannsen and Röder, 2014*; *Lewkowicz, 2014*). These functionally tuned networks (including the experience dependent feedback connectivity) constitute models of the sensory world (*Fiser et al., 2010*). Their elaboration might go together with a switch towards model-based learning which is characterized by a larger context dependency. As some parts of the neural networks stabilize, learning must partially involve additional neural systems to guarantee that the adaptations necessary throughout life are realized without risking the loss of essential crossmodal knowledge. For example, prism wearing during the sensitive phase has been reported to change the connectivity between the central (ICC) and external (ICX) inferior colliculus of the auditory midbrain of barn owls. By contrast, crossmodal adaptation to prisms later in the critical period seems to be mediated by a reorganization of the optical tectum to which the ICX projects (*Knudsen, 2002*). Moreover, *Bergan et al. (2005)* reported that crossmodal adaptions to prims were enhanced in adult owls when they were allowed to hunt, that is, when such adaptations were particular needed. In accord with these findings in owls, we demonstrated that adult learning of crossmodal combinations depended on task relevance (Experiment 2b). Thus, as suggested by *Keuroghlian and Knudsen (2007)* and *Bavelier et al. (2010)*, neuroplasticity in adults seem to require to a larger extend attention and behavioral relevance and thus the involvement of additional higher order neural systems. Task relevance or attention constitute specific top-down influences on sensory representations and are thus mediated via the feedback connections which become progressively tuned and elaborated during development (*Dehaene-Lambertz and Spelke, 2015*).

The present study was able to dissociate the processes for the learning of probabilities of sensory elements and for the learning of conditional probabilities of the sensory elements comprising crossmodal stimuli. All groups were sensitive to 'Rare deviant stimuli'. To detect 'Rare deviant stimuli' the frequency of sensory elements rather than conditional probabilities had to be traced. Indeed, it was possible to detect 'Rare deviant stimuli' only based on one of the two sensory elements. The ERP effect to 'Rare deviant stimuli' started earlier than the ERP effects to 'Rare recombined stimuli'. Such early deviant effects are typical for an auditory mismatch negativity (MMN, see *Schröger and Wolff, 1996*; *Cheour et al., 2000*). Therefore, we suggest that the observed 'Rare deviant stimuli' effect, similarly as has been proposed for the MMN reflects, indicates a sensory memory trace, which represents the frequency of sensory elements (*Näätänen and Alho, 1995*). By contrast, the detection of conditional probabilities of crossmodal stimuli cannot be based on such (unisensory) sensory memory traces. Thus, we speculate that the detection of 'Rare deviant stimuli', is based on modality specific systems (*Frost et al., 2015*). Although it has been reported that auditory mismatch responses are enhanced by redundant crossmodal (somatosensory) information in adults such a multisensory enhancement was only observed for later time epochs (>200 ms; *Butler et al., 2012*) than the first 'Rare deviant stimulus' effect of the present study.

Since we argue that the change in learning mode during development is related to functional specialization, the strong lateralization of both ERP effects in infants seems rather surprising. The differentiation of 'Frequent standard stimuli' and 'Rare recombined stimuli' requires the detection of conditional probabilities. This ability has been postulated as a precursor of language learning (*Saffran et al., 1996*). Indeed, it has been shown with structural imaging techniques that many hemispheric asymmetries, in particular those related to the language system (*Friederici, 2009*), exist at birth or shortly thereafter (see *Dehaene-Lambertz and Spelke, 2015*). Thus, we speculate that the strong left lateralized ERP difference between 'Frequent standard stimuli' and 'Rare recombined stimuli' might reflects a recruitment of similar neural circuits that have been proposed to enable the detection of word boundaries (*Saffran et al., 1996*), non-adjacent transitional probabilities and possibly syntactical rules (*Friederici, 2002*; *Friederici et al., 2006*). Thus, this neural system might, partially independently of sensory modality and domain, allow for detecting any type of statistical relations (*Kuhl, 2010*; *Aslin and Newport, 2014*). In fact, a correlation of syntactic competence and statistical learning skills in children has been reported (*Kidd and Arciuli, 2016*). The right lateralized ERP effect to 'Rare deviant stimuli' was not unique to the infant group, but was as well observed in the adults tested with the same passive design (Experiment 1b). Interestingly such a lateralization was neither found for Experiment 2a nor for Experiment 2b, in which the adult participants were actively engaged in a task. We speculate that 'Rare deviant stimuli' elicited a reflexive and

exogenous attention shift to the rare sensory features. Such reflexive spatial attention orienting has often been associated with right parietal brain regions (*Okada et al., 2008*; *Mort et al., 2003*; *Chica et al., 2011*). In contrast, in Experiment 2a and 2b, participants had to allocate attention to a specific stimulus or stimulus combination and it was adaptive to avoid exogenous attention shifts.

In the present study ERP effects in adults were of different polarity and had a shorter latency compared to the infant group. We linked ERP effects in infants and adults based on the experimental manipulations and their relative timing. Due to the immature brain (e.g. incomplete myelination) of infants and children it is a common finding that absolute latencies of ERP effects are longer in the developing brain. Moreover, it has repeatedly been reported that polarities of effects differ in infancy or children and adulthood (*Kouider et al., 2015*; *Neville et al., 2013*; *Nelson, 1997*; *de Haan and Halit, 2001*).

In conclusion, our study demonstrates that six-month old infants were able to quickly learn crossmodal statistics through a mere passive exposure, whereas adults learned the same crossmodal combinations only when they were task relevant. Thus, we provide first evidence for a higher sensitivity for crossmodal statistics in infants compared to adults, indicating age-dependent mechanisms for the learning of arbitrary crossmodal combinations. We speculate that initial passive association learning allows infants to quickly form first internal models of their sensory environment. In adulthood these internal models are adjusted if this is behavioral adaptive.

# Materials and methods

## Experiment 1

### Participants: Experiment 1a
Sixty-two six-month-old infants (±10 days) took part. Infants were recruited from the local registration offices. All participating infants were born full-term (38–41 weeks), had a typical prenatal and perinatal history and no known neurological or developmental problems. Parents gave their written consent and were informed about their right to abort the experiment at any time. They received a small present for their children (toy or picture book) for taking part. Thirty-three participants were excluded from the analyses because of too many artifacts in the EEG recordings, leaving a total of twenty-nine data sets for the final statistical analyses (17 female, 12 male). Note that an exclusion rate of approximately 50% due to artifacts is not uncommon in infant research (*DeBoer et al., 2007*). The study (including Experiment 1a and 1b) was performed in accordance with the ethical standards laid down in the Declaration of Helsinki in 1964. The procedure was approved by the ethics board of the German Psychological Society (DGPs).

### Stimuli and design: Experiment 1a
The experiment comprised three auditory and three visual stimuli, combined into crossmodal pairs of one visual and one auditory stimulus. The visual and auditory stimuli were always simultaneously presented. All three auditory stimuli had the equal loudness but differed in sound frequency (400, 1000 or 1600 Hz); they were presented for 500 ms each via two loudspeakers. The visual stimuli consisted of three geometric shapes (circle, triangle, and square; size: 10°) combined with three different colors (green, red, and blue) and were presented in the middle of a computer screen for 500 ms.

Participants were exposed to two frequently occurring audio-visual standard combinations (A1V1, A2V2, each with p=0.35, 'Frequent standard stimuli') and three infrequently occurring audio-visual deviant combinations. The latter consisted of (1) two rare recombinations of the auditory and visual stimuli comprising the 'Standard stimuli' (A1V2, A2V1, each with p=0.10, 'Rare recombined stimuli') and (2) one rare audio-visual combination of a deviant auditory and a deviant visual stimulus (A3V3, p=0.10, 'Rare deviant stimuli'), not occurring in the combinations of the 'Frequent standard stimuli' and the recombined stimuli. Due to the recombining of the auditory and visual elements of the 'Frequent standard stimuli', the likelihood of the auditory and visual elements comprising the 'Frequent Standard stimuli' and the 'Rare recombined stimuli' were identical. By contrast, 'Rare deviant stimuli' consisted of auditory and visual elements, which had an overall lower likelihood. Thus, it was possible to distinguish processes related to the likelihood of sensory elements ('Frequent standard stimuli' vs. 'Rare deviant stimuli') and processes related to the detection of crossmodal combinations ('Frequent standard stimuli' vs. 'Rare recombined stimuli').

The inter stimulus interval between the different crossmodal stimuli amounted to 1500 ms. The visual and auditory stimuli used for each crossmodal condition was consistently for each participant but was counterbalanced over participants. The experiment was divided into five experimental blocks, each comprising 60 trials resulting in a total of 300 trials. For each block the proportion of the three conditions was 70: 20: 10% (see *Table 3*). Thus, even if the experiment was prematurely aborted, each infant received the correct ratio of stimuli.

## Procedure: Experiment 1a

Experiment 1a took place in a sound-attenuated and electrically shielded room. During the experiment, the infants sat on their parents' laps. The computer screen, displaying the visual stimuli, was positioned on a table at a distance of approximately 60 cm from the participants. Infants' heads were aligned with the center of the screen. The two loud speakers were positioned behind the computer screen.

To make sure that the infants attentively observed the stimuli, a black and white video was continuously played in the background. This video consisted of 30 different sequences of centrally moving patterns, e.g. randomly moving stars or flying balloons focusing the viewing direction to the center of the computer screen. All sequences were ten seconds long and were presented without intermediate breaks. To control whether the infants were actually looking at the computer screen when the experimental visual stimuli were presented, a small camera, placed on top of the computer screen, recorded the infants' heads. The camera was connected to the EEG recording computer to enable a continuous control of the child's attention as well as the EEG signal during the course of the experiment. If the infant did not look at the screen during the presentation of the stimuli, a marker was manually inserted by the experimenter in the EEG data file and the associated EEG segments were later taken out of the analysis. To avoid interfering signals, parents were instructed not to talk to their children during the time the EEG was recorded. Whenever the infant showed signs of discomfort or restlessness, the experiment was paused. Occasionally, a hand puppet was used during such breaks to keep the infants alert and to make sure that they attended to the computer screen when the experiment was continued. The EEG recording only continued if both the child and the parent were content. The testing time for all infants ranged between five and ten minutes (M = 7.2 min, SD = 1.6). Together with the preparation time, the infants and their parents spent approximately forty-five minutes in the laboratory.

## Electrophysiological recording and data analyses: Experiment 1a

EEG data were collected from 45 scalp sites using active Ag/AgCl electrodes (Brain Products, Easy-cap GmBH, Herrsching) mounted in an elastic cap (Electro Cap International, Inc.). The electrodes were placed according to the international 10–10 system (see *Figure 5*). EEG Data were recorded continuously using a band-pass filter of 0.01–250 with a sampling rate of 500 Hz (Brain Products, Munich, Germany). The electrode FPz served as online reference electrode and the ground electrode was applied at AF3. Data were re-referenced offline to the average of the recordings of electrodes TP9 and TP10, which are located close to the mastoids. Artifacts were rejected manually after visual inspection of each individual EEG trial. Trials with artifacts such as head movements, eye blinks, eye movements or electrical noises were removed from further analyses. The first 15 trials of each dataset were excluded since the participants were not yet familiarized with the relative proportions of each stimulus condition. Noisy channels were interpolated by calculating the average of the four adjacent electrodes (*Picton et al., 2000*). On average, three electrodes were interpolated for

**Table 3.** Experimental design of Experiment 1a and Experiment 1b.

| Stimuli | Proportion | | Condition (number of trials) |
|---|---|---|---|
| Auditory 1 – Visual 1 (A1V1) | 0.35 ⎫ | | Frequent standard stimuli (210) |
| Auditory 2 – Visual 2 (A2V2) | 0.35 ⎬ 0.70 | | |
| Auditory 1 – Visual 2 (A1V2) | 0.10 ⎫ | | Rare recombined stimuli (60) |
| Auditory 2 – Visual 1 (A2V2) | 0.10 ⎬ 0.20 | | |
| Auditory 3 – Visual 3 (A3V3) | 0.10 ⎫ 0.10 | | Rare deviant stimuli (30) |

DOI: https://doi.org/10.7554/eLife.28166.009

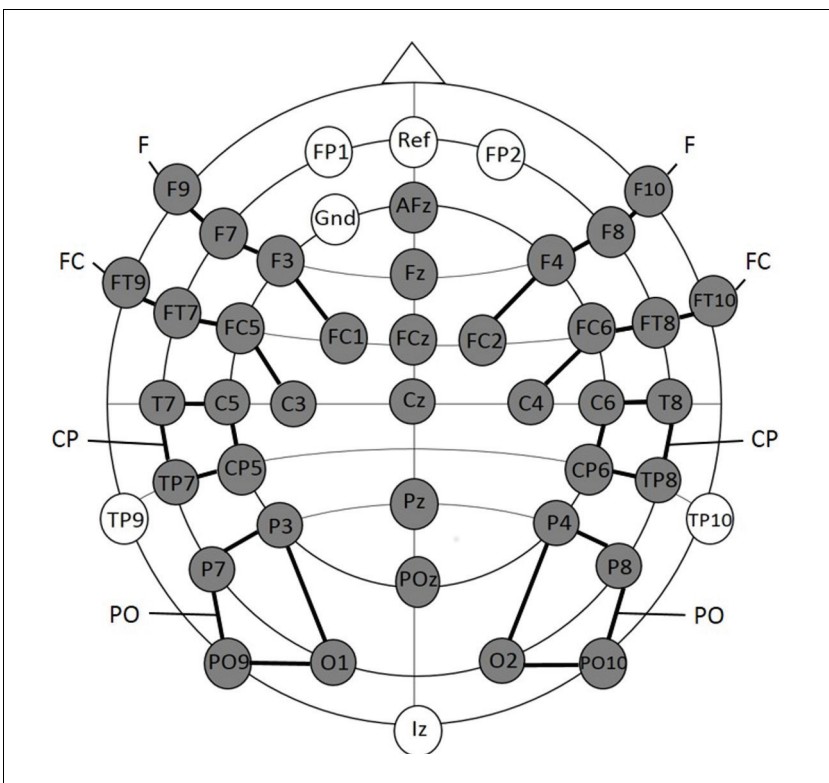

**Figure 5.** Electrode placement for experiment 1a. The grey electrodes were included in the statistical analyses. Clusters are indicated by black connecting lines and were named according to their location along the anterior-posterior axis.

DOI: https://doi.org/10.7554/eLife.28166.010

each participant. EEG data sets of infants (n = 21) comprising less than 10 trials per condition were excluded from the final statistical analyses (see participants Experiment 1a). For the statistical analyses, the lateral electrodes were grouped into four clusters for each hemisphere; each cluster comprised four electrodes (see *Figure 5*): the left hemisphere: (1) Frontal (F): F9, F7, F3, FC1; (2) Fronto-central (FC): FT9, FT7, FC5, C3; (3) Central-parietal (CP): T7, C5, TP7, CP5; (4) Parietal-occipital (PO): P3, P7, PO9, O1 and the right hemisphere: (1) Frontal (F): F10, F8, F4, FC2; (2) Fronto-central (FC): FT10, FT8, FC6, C4; (3) Central-parietal (CP): T8, C6, TP8, CP6; (4) Parietal-occipital (PO): P4, P8, PO10, O2. The midline electrodes AFz, Fz, FCz, Cz, Pz, and POz were separately analyzed. EEG data were segmented into epochs from 100 ms pre-stimulus to 1100 ms post-stimulus onset. Epochs were baseline corrected by means of the 100 ms pre-stimulus interval.

Mean amplitudes were calculated separately for each condition and participant for the following time epochs based on visual inspection of the group mean average: (1) 200–420 ms and (2) 420–1000 ms. To evaluate differences between conditions, a repeated measurement ANOVA comprising the within subject factors *Condition* (three levels: 'Frequent standard stimuli' vs. 'Rare recombined stimuli' vs. 'Rare deviant stimuli'), *Hemisphere* (two levels: left vs. right) and *Cluster* (four levels: F vs. FC vs. CP vs. PO) was calculated separately for each of the two time windows.

Significant interactions including the factor *Condition* were followed up with sub-ANOVAs, calculated separately for each cluster. Significant main effects of *Condition* or interactions of *Condition* and *Hemisphere* were further analyzed with paired t-tests: (1) 'Frequent standard stimuli' vs. 'Rare deviant stimuli' and (2) 'Frequent standard stimuli' vs. 'Rare recombined stimuli'. The midline electrodes were separately analyzed with an ANOVA comprising the factors *Condition* (three levels: Standard vs. New Combination vs. New Stimuli) and *Electrode* (six levels: AFz vs. Fz vs. Cz vs. Pz vs. POz). Similar to the cluster analysis, significant interactions between the factor *Condition* and *Electrode* were further analyzed by calculating sub ANOVAs and paired t-tests separately for each electrode. The Huynh-Feldt correction was applied to all analyses comprising within subject factors with

more than two levels. To correct for multiple comparisons, p-values of the t-tests were adjusted with the Bonferroni-Holm method. Only main effects and interactions, including the factor *Condition,* as well as significant post hoc tests are reported.

### Participants: Experiment 1b

Twenty-seven young adults recruited from a student-subject database of the Institute for Psychology (University of Hamburg) were tested. They received either 8 €/hour or course-credit. All participants had normal or corrected-to-normal vision, normal hearing and were free of neurological problems. All participants gave their informed consent. Four participants were excluded from the analysis due of too many artifacts in the EEG. A total of twenty-three participants were included in the final analyses (11 male, mean age 23.5 years, range 19–31)

### Stimuli and design: Experiment 1b

The stimuli and experimental design of Experiment 1b were identical to Experiment 1a (see *Table 3*).

### Procedure: Experiment 1b

Experiment 1b took place in the adult EEG lab of the Biological Psychology and Neuropsychology section of the University of Hamburg. It was constructed by the same company as the Baby lab and had the same light sources, sound attenuating, and electrical shielding system. The experimental room was dimly lit and the participants were seated in a comfortable chair in front of a table. All devices used were the same as for Experiment 1a. The computer screen, displaying the visual stimuli and background video, was positioned at eye level on a table at a distance of approximately 60 cm from the participants (size of the visual stimuli: 7°). The two loud speakers were located behind the computer screen. Before the experiment started, participants received written instructions concerning the procedure of the experiment. In addition, they were asked to sit as still as possible, to limit their eye blinking during the recording of the experimental blocks and to continuously look at the fixation point. To control that the participants attended to the computer screen participants' heads were recorded via a small camera, placed on top of the computer screen, during the experiment.

### Electrophysiological recording and data analyses: Experiment 1b

EEG recording and data analyses were identical to Experiment 2a and 2b. Note, that the similar results for the ERPs to 'Rare deviant stimuli' in infants and adults, including the lateralization, exclude the possibility that differences in analyzing procedures contributed to the below reported other group differences.

## Experiment 2

### Participants

Seventeen healthy university students took part in the experiment. The participants were recruited from a student-subject database of the Institute of Psychology at the University of Hamburg. They received either 8 €/hour or course-credit. All participants had normal or corrected-to-normal vision, normal hearing and no neurological problems. Five participants were excluded from the analysis due to too many artifacts in the EEG or insufficient task performance (less than 70% correct target detection), leaving a total of twelve participants for the final analyses (four male, age 20–31 years, mean = 23.8 years). All participants gave their informed consent. The study was performed in accordance with the ethical standards laid down in the Declaration of Helsinki in 1964. The procedure was approved by the ethics board of the German Psychological Society (DGPs).

### Stimuli and design

The design of Experiment 2 was similar to Experiment 1, but the stimuli and the experimental setting was adjusted. A visual LED was located inside a small wooden front (22 × 24 cm) which was covered with a black cloth. The wooden front was placed on top of a black box, to make sure that the position of the LED was at eye-level at a distance of approximately 85 cm from the participants. The LED was activated for 100 ms in four possible colors: red, blue, green or yellow. Auditory stimuli (400, 800, or 1600 Hz) were presented for 100 ms via two speakers which were positioned adjacent to the

wooden front. Crossmodal stimuli were made by combining one of the sounds with one of the LED colors. Crossmodal combinations were counterbalanced over conditions and participants. In contrast to Experiment 1b, adults were engaged in a task and had to detect a target stimulus rather than being passively exposed to a sequence of crossmodal stimuli. The target stimulus was either unrelated to the crossmodal combinations (Experiment 2a) or addressed a specific crossmodal combinations (Experiment 2b), resulting in two different experiments.

In Experiment 2a the 'Frequent standard stimuli' (A1V1, A2V2) were presented with a probability of p=0.30 each while the 'Rare recombined stimuli' (A1V2, A2V1) and 'Rare deviant stimuli' (A3V3) had a probability of p=0.10 each. An additional unimodal visual stimulus (p=0.10, V4) served as target stimulus (see *Table 4A*). We used an additional unimodal stimulus as target to guarantee that participants were attending the stimuli. A visual rather than an auditory or crossmodal stimulus was used as target stimulus to guarantee that participants did not close their eye and to render crossmodal stimuli totally task irrelevant in Experiment 2a.

In Experiment 2b no unimodal V4 was included, but one of the 'Rare recombined stimuli' (either A1V2 or A2V1) was defined as the target stimulus rendering crossmodal combinations task relevant. A1V1 and A2V2 were presented with a probability of p=0.35 each while the probability for A1V2, A2V1, and A3V3 was p=0.10 each (see *Table 4B*). All participants took part in both experiments. The order of the two experiments as well as the specific audio-visual combinations used for the different conditions were counterbalanced over participants. However, the assignment of auditory-visual combinations to conditions was kept the same for each participant in Experiment 2a and 2b. Stimuli were presented in six blocks with 200 trials per block.

## Procedure

The experiment took place in a dimly lit, sound-attenuating, and electrical shielded room. The participants were seated in a comfortable chair at a table approximately 85 cm from the box that contained the visual LED. The target stimulus was presented three times prior to the start of the experiment, to allow participants to get acquainted with the target. Responses to the target stimuli were made by means of a custom made button box, placed near the dominant hand. Participants were instructed to sit as still as possible and to keep their eyes focused on the LED. Experiment 2a and 2b lasted for twenty to thirty minutes each (including breaks). The total testing time, which included briefing of the participant, practice trails and EEG application, was approximately 1 hr and 45 min for both experiments.

## Behavioral analysis

All button presses within 100 and 1000 ms following stimulus presentation were considered as valid responses. Hit, miss and false alarm rates were calculated and average reaction times to targets were derived for both Experiment 2a and 2b.

**Table 4.** Experimental design of (A) Experiment 2a and (B) Experiment 2b.

| A | Stimuli | Proportion | | Condition (number of trials) |
|---|---|---|---|---|
| | Auditory 1 – Visual 1 (A1V1) | 0.30 | 0.60 | |
| | Auditory 2 – Visual 2 (A2V2) | 0.30 | | Frequent standard stimuli (720) |
| | Auditory 1 – Visual 2 (A1V2) | 0.10 | 0.20 | Rare recombined stimuli (240) |
| | Auditory 2 – Visual 1 (A2V2 | 0.10 | | |
| | Auditory 3 – Visual 3 (A3V3) | 0.10 } 0.10 | | Rare deviant stimuli (120) |
| | Visual 4 | 0.10 } 0.10 | | Unimodal target stimuli (120) |
| B | Stimuli | Proportion | | Condition (number of trials) |
| | Auditory 1 – Visual 1 (A1V1) | 0.35 | 0.70 | |
| | Auditory 2 – Visual 2 (A2V2) | 0.35 | | Frequent standard stimuli (840) |
| | Auditory 1 – Visual 2 (A1V2) | 0.10} 0.10 | | Rare recombined stimuli (120)/ |
| | Auditory 2 – Visual 1 (A2V2 | 0.10} 0.10 | | Target stimuli (120) |
| | Auditory 3 – Visual 3 (A3V3) | 0.10 } 0.10 | | Rare deviant stimuli (120) |

DOI: https://doi.org/10.7554/eLife.28166.011

## Electrophysiological recording and data analysis

EEG data were collected from 74 scalp sites using active Ag/AgCl electrodes (Brain Products, Easycap GmBH, Herrsching) mounted on an elastic cap (Electro Cap International, Inc.). Data were recorded continuously using a band-pass filter of 0.01–250 with a sampling rate of 500 Hz (Brain Products, Munich, Germany). The electrodes were placed according to the international 10–10 system (see *Figure 6*). One additional electrode was positioned below the left eye to record vertical eye movements. A left earlobe electrode served as online reference electrode. EEG data were filtered offline with a low-pass filter with a 40 Hz cut-off and were re-referenced offline to an average reference. Electrodes positioned close to the outer canthi of each eye (F9 and F10) served for recording horizontal eye movements. An independent component analysis (ICA) was run for each EEG data set, which defined 30 time-independent components representing the data (Makeig, Debener, Onton & Delorme, 2004). Components representing artifacts such as eye blinks, eye movements, electrical noise or heart beat were manually detected and rejected from further analyses. The first 75 trials (Experiment 2a and 2b) or the first 15 trials (Experiment 1b) of each dataset were excluded since the participants were not yet familiarized with the relative proportions of each stimulus condition. The lateral electrodes were grouped into six clusters for each hemisphere; each cluster comprised five electrodes (see *Figure 6*): (1) Frontal (F): F1, F3, F5, F7, F9 (2) Fronto-central (FC): FC1, FC3, FC5, FT7, FT9 (3) Central (C): C1, C3, C5, T7 (4) Centro-parietal (CP): CP1, CP3, CP5, TP7, TP9 (5) Parietal (P): P1, P3, P5, P7, P9 (6) Parieto-occipital (PO): PO3, PO7, PO9, O1, O9) and for the right hemisphere: (1) Frontal (F): F2, F4, F6, F8, F10 (2) Fronto-central (FC): FC2, FC4, FC6, FT8, FT10 (3) Central(C): C2, C4, C6, T8 (4) Centro-parietal (CP): CP2, CP4, CP6, TP8, TP10 (5) Parietal (P): P2, P4, P6, P8, P10 (6) Parieto-occipital (PO): PO4, PO8, PO10, O2, O10). The midline electrodes Fz, FCz, Cz, CPz, Pz, POz, and Oz were separately analyzed. EEG data were segmented into epochs starting 100 ms before the stimulus onset and lasting for 1000 ms post stimulus onset. Epochs were baseline corrected with a pre-stimulus interval of 100 ms. Mean amplitudes were

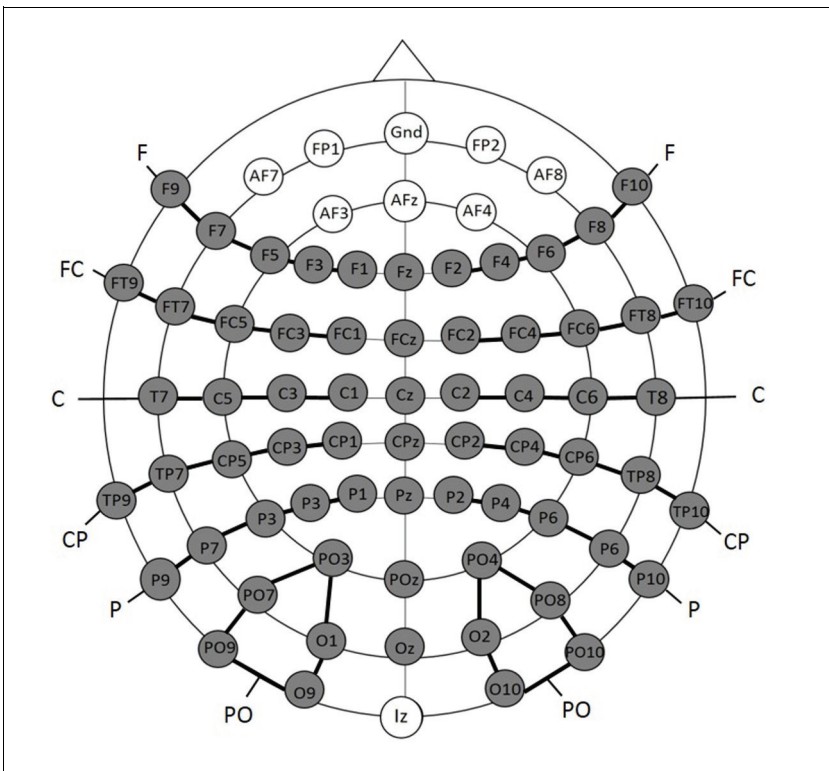

**Figure 6.** Electrode placement for Experiment 2a and 2b; the grey electrodes were included in the statistical analyses. Clusters are indicated by black connecting lines and were named according to their location along the anterior-posterior axis.

DOI: https://doi.org/10.7554/eLife.28166.012

calculated separately for each condition and participant for the following time epochs based on visual inspection of the group mean average: (1) 80–160 ms and (2) 250–850 ms. The statistical analyses were the same as described for Experiment 1a.

## Acknowledgements

This work was supported by the European Research Council (ERC-2009-AdG 249425 CriticalBrainChanges) and grant 'Crossmodal Learning' of the City of Hamburg. We thank Rebecca Nixdorf for help with data acquisition and József Fiser and Erich Schröger for comments and suggestions. We are particularly grateful to the parents and their children for taking part.

## Additional information

### Funding

| Funder | Grant reference number | Author |
| --- | --- | --- |
| Horizon 2020 | ERC-2009-AdG 249425 CriticalBrainChanges | Brigitte Röder |
| City of Hamburg | Crossmodal Learning | Brigitte Röder |

The funders had no role in study design, data collection and interpretation, or the decision to submit the work for publication.

### Author contributions

Sophie Rohlf, Conceptualization, Data curation, Software, Formal analysis, Visualization, Writing—original draft, Project administration; Boukje Habets, Conceptualization, Data curation, Software, Formal analysis, Project administration, Writing—review and editing; Marco von Frieling, Conceptualization, Data curation, Project administration, Writing—review and editing; Brigitte Röder, Conceptualization, Resources, Formal analysis, Supervision, Funding acquisition, Writing—original draft, Project administration

### Author ORCIDs

Sophie Rohlf http://orcid.org/0000-0002-8947-5613

### Ethics

Human subjects: Parents (Experiment 1a) and participants (Experiment 1b/2a/2b) gave their written consent and were informed about their right to abort the experiment at any time. All experiments were performed in accordance with the ethical standards laid down in the Declaration of Helsinki in 1964. The procedure was approved by the ethics board of the German Psychological Society (DGPs).

### Decision letter and Author response

Decision letter https://doi.org/10.7554/eLife.28166.014
Author response https://doi.org/10.7554/eLife.28166.015

## Additional files

### Supplementary files

• Transparent reporting form
DOI: https://doi.org/10.7554/eLife.28166.013

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
