## [Decision Letter]

Thank you for submitting your article "Infants are superior in implicit crossmodal learning and use other learning mechanisms than adults" for consideration by *eLife*. Your article has been reviewed by three peer reviewers, and the evaluation has been overseen by a Reviewing Editor and Sabine Kastner as the Senior Editor. The reviewers have opted to remain anonymous.

The reviewers have discussed the reviews with one another and the Reviewing Editor has drafted this decision to help you prepare a revised submission.

Summary:

Overall the three reviewers are positive on the experimental design and the interesting findings. However, each reviewer has concerns about the manuscript many of which concern the analyses of the ERP data but also the theoretical framing and interpretation of the Results section.

Essential revisions:

Each reviewer raises largely unique concerns but that are convergent on (a) the soundness of selection criteria for the ERP analyses (reviewer #1: time-window selection; reviewer #3: ROIs, baseline); (b) the justification and power for the adult ERP studies (reviewer #2); (c) the interpretation and theoretical framing of largely null findings in the adult ERPs (reviewer #1 and #2). Reviewers #2 and #3 also provide suggestions for rewriting and reorganizing the manuscript that will aid the reader and also make the manuscript more suitable for publication in *eLife*.

*Reviewer #1*:

The authors present 4 experiments (3 adults, 1 infants) examining statistical learning of audio-visual stimuli in infants and adults. Their experimental design allows them to examine both learning for the basic frequency of audiovisual stimuli (standard vs. deviant stimuli) and also the recombinations of audiovisual stimuli (standard vs. recombined stimuli). They find evidence of robust and consistent ERP signatures of the former type of learning across ages and experiments. However, while they find evidence of the latter type of learning in infants in a single experiment, they find evidence of differential neural responses across standard vs. recombined stimuli only in the last experiment in adults. Overall, the study is very interesting and a timely and theoretically-central topic. However, the results particularly for the emergence of this audiovisual combination learning in adults is weaker without clear justification for why this type of learning is similar to what is being demonstrated in infants. Moreover, the theoretical explanation of the specific pattern of phenomena reported is at times not explicitly reasoned and at others is not clearly consistent, even within the paper.

Reconcile the argument for better infant learning in this paper with evidence that, particularly for visual SL, there are well-documented increased in SL abilities across childhood (e.g. Arciuli & Simpson, 2011). In the Discussion section, the authors do mention that some studies have found better learning in older than younger children and that it could be that greater complexity is better learned at older ages. This also runs contrary to the current findings where it is the more complex aspect of learning that is better in infants than adults.

The authors present an argument that learning would be more passive in infants and more active or task-relevant in adults. However, their hypotheses are really about learning standard vs. deviant as opposed to crossmodal associations. The referenced theoretical paper from the Knudsen group is simply about auditory learning and doesn't make assumptions about cross-modal learning per se. Why would the authors hypothesize this dissociation between passive vs. task-relevant learning across these different types of learning (standard/deviant VS. cross-modal associations)?

Feedback vs. feedforward. If standard vs. deviants are more consistent with feedforward connections and crossmodal associations are more consistent with feedback. It could be evidence of greater feedback, crossmodal connectivity in infants. This is very indirect evidence (the authors are clear that it is speculative) but it is not clear how it is consistent with the authors claim that feedback connections are better tuned later in development (as per Dehaene-Lambertz & Spelke, 2015) and thus would support less learning. It is not clear how these two views are consistent especially given that the cross-modal associations are learnable when they are task-relevant: Task-relevance presumably doesn't create broader feedback connectivity.

Analyses:

Visual inspection was used to select different time-windows for analysis even across experiments where the stimuli were identical (Experiments 2A and 2B) and indeed even the participants were identical. Is there an independent justification for selecting different time-windows beyond visual inspection? Would the same results obtain with identical time-windows across the two experiments?

Differences between frequent vs. deviant-recombined stimuli were with a more negative response in infants that is late vs. a more positive response in adults. why do the authors conclude that these are equivalent or evidence of a similar learning mechanism?

Discussion section:

Early discrimination of standard vs. deviant stimuli. Consider in relation to Kouider et al., 2015 where early components were modulated by cross-modal associations.

*Reviewer #2:*

The authors report that 6-month-old infants rapidly learn cross-modal associations implicitly, while in adults there is no evidence of such learning unless the particular associations are explicitly attended to. This is an impressive study with interesting and important results which add to our knowledge of the organisation of human multisensory processing and learning. I have only minor comments, generally related to structure and presentation.

The plan of experiments and results is rather hard to follow and keep in mind. We get methods for multiple experiments using slightly different setups, then a lot of different results (which are not interpreted/summarised in an interim way), before discussion. The rationale for the specific design (why we have those particular three conditions) is nowhere spelled out either. It is a clever design, and a good choice by the authors, but the authors should take some time to explain the logic behind their choice (why these conditions) in the Introduction or Materials and methods section. E.g. there is clearly logic behind matching the (low) frequencies of the "rare recombined" and "deviant" stimuli but the point of this manipulation is never explicitly explained or discussed.

I would suggest a few things to make the paper more accessible. (1) presenting the experiments as a sequence – with 2A and 2B as follow-ups after interim conclusions based on Experiment 1, rather than the (not very plausible to me) suggestion in intro that they were all planned and predicted from the outset, as if Experiments 2A and 2B were run without knowing any results from Experiment 1. (3) as suggested above, explaining rationale for the choice of different conditions and how they are matched (e.g. in terms of probability) more clearly when these are introduced. (2) Within the Results section, add some interim summaries of what each set of results/analyses has shown ("In summary…"). (6) Consider some kind of table or other way of summarising the results of the different analyses. I found myself writing the 3 condition names (FS / RR / RD) for each analysis and joining these with lines to show where some differed – one easily visualisable outcome from this is that many clusters / epochs / ages show a FS-RD difference whereas the FS-RR difference (of most interest) is much rarer. A high-level table or visualisation that makes this clear would be helpful.

My other substantive comment is on power of the adult vs infant analysis to find the effects of interest – this is mentioned in the Discussion section, but the argument would be more convincing with a quantitative treatment (e.g. comparing the size of effect that was detected successfully with adults with the size of effect that was seen in infants but not detected in adults).

Reviewer #3:

The authors show that infants can learn a cross-modal predictive mapping implicitly while adults can only learn when the mapping is task-relevant, demonstrating superior cross-modal plasticity in infants.

This is potentially an interesting finding although the paper does not strike me as being optimally written for *eLife*. The paper is very long, structured like a classical experimental psychology paper and contains many details, which is not the standard for this kind of papers in *eLife*. I would strongly recommend shortening it and maybe presenting it as a brief article because at the end the main message is lost in many details which are much less relevant.

Another major issue is the design and analysis of the data. As I understand it, the baseline correction is made out of the period just preceding the visual stimulus. This would mean that it includes auditory potentials. The baseline should be subtracted in the period before the whole pair of stimuli, otherwise the visual ERPs will necessarily be contaminated by the (inverse of) the auditory potentials. This might fully explain the effect induced by the rare deviant stimuli condition, where there was also a new auditory stimulus (A3).

Another issue is about the region of interest approach. Because the analysis is made on the visual stimuli, one might expect a focus on visual electrodes and their modulation as a function of condition. For infants, the cluster CP seems to show an effect, but it does not appear on the figures of the ERPs.

---

## [Author Response]

Reviewer #1:The authors present 4 experiments (3 adults, 1 infants) examining statistical learning of audio-visual stimuli in infants and adults. Their experimental design allows them to examine both learning for the basic frequency of audiovisual stimuli (standard vs. deviant stimuli) and also the recombinations of audiovisual stimuli (standard vs. recombined stimuli). They find evidence of robust and consistent ERP signatures of the former type of learning across ages and experiments. However, while they find evidence of the latter type of learning in infants in a single experiment, they find evidence of differential neural responses across standard vs. recombined stimuli only in the last experiment in adults. Overall, the study is very interesting and a timely and theoretically-central topic. However, the results particularly for the emergence of this audiovisual combination learning in adults is weaker without clear justification for why this type of learning is similar to what is being demonstrated in infants. Moreover, the theoretical explanation of the specific pattern of phenomena reported is at times not explicitly reasoned and at others is not clearly consistent, even within the paper.Reconcile the argument for better infant learning in this paper with evidence that, particularly for visual SL, there are well-documented increased in SL abilities across childhood (e.g., Arciuli & Simpson, 2011). In the Discussion section, the authors do mention that some studies have found better learning in older than younger children and that it could be that greater complexity is better learned at older ages. This also runs contrary to the current findings where it is the more complex aspect of learning that is better in infants than adults.

Rare deviant stimuli’ were in fact sensory deviants (both the auditory and visual elements had a lower likelihood compared to all other sensory elements) and could be detected based on one sensory element alone. In the revised manuscript we discuss this idea in more detail. We link the ‘Rare deviant stimuli’ effect to a typical mismatch negativity (MMN) which was related to a sensory memory trace. We discuss that likely intramodal processing was sufficient to detect ‘Rare deviant stimuli’ (Discussion section). By contrast, ‘Rare recombined stimuli’ could only be detected based on the crossmodal statistics which involved knowledge of conditional probabilities of the auditory and visual elements. We consider both processes as qualitatively distinct rather than more or less complex. Actually, the crossmodal statistics or conditional probabilities were relatively simple compared to e.g. the visual and auditory sequences used by Aciuli & Simpson, 2011. Thus, more complex rules and a higher number of possible sequences seem to be better learned the older the children are. Working memory (WM) may be an involved factor. Simple crossmodal associations of two auditory and two visual stimuli as used in the present study unlikely touch the limits of WM. This discussion has been added to the Discussion section.

The authors present an argument that learning would be more passive in infants and more active or task-relevant in adults. However, their hypotheses are really about learning standard vs. deviant as opposed to crossmodal associations. The referenced theoretical paper from the Knudsen group is simply about auditory learning and doesn't make assumptions about cross-modal learning per se. Why would the authors hypothesize this dissociation between passive vs. task-relevant learning across these different types of learning (standard/deviant VS. cross-modal associations)?

Detecting ‘Rare deviant stimuli’ required that participants kept track of the probability of sensory elements, while detecting ‘Rare recombined stimuli’ involved the learning of conditional probabilities across sensory systems. An unimodal experiment with an analogous design as employed in the present study would be necessary to test whether the enhanced sensitivity of infants can be observed for simple conditional probabilities in general or whether it is specific for crossmodal conditional probabilities. The Knudsen lab indeed provided evidence that task relevance enhances crossmodal learning. (Bergan et al., 2005). Though this study on prism adaptation is cited in Keuroghlian & Knudsen, 2007 (Figure 9), we explicitly refer to Bergan et al., 2005 in the revised manuscript (Discussion section).

Feedback vs. feedforward. If standard vs. deviants are more consistent with feedforward connections and crossmodal associations are more consistent with feedback. It could be evidence of greater feedback, crossmodal connectivity in infants. This is very indirect evidence (the authors are clear that it is speculative) but it is not clear how it is consistent with the authors claim that feedback connections are better tuned later in development (as per Dehaene-Lambertz & Spelke, 2015) and thus would support less learning. It is not clear how these two views are consistent especially given that the cross-modal associations are learnable when they are task-relevant: Task-relevance presumably doesn't create broader feedback connectivity.

This part of the discussion has been majorly revised. We first discuss how the maturation of crossmodal connectivity including feedback connectivity might be related to the age-dependent learning effects observed in the present and some previous (unimodal) studies (Discussion section). In the consecutive paragraph we discuss how detecting ‘Rare recombined stimuli’ might differ from detecting ‘Rare deviant stimuli’ (see 1.1 and 1.2 as well) (Discussion section). In the revised manuscript we do not associate the detection of the ‘Rare deviant stimuli’ with feedforward connectivity.

Analyses:Visual inspection was used to select different time-windows for analysis even across experiments where the stimuli were identical (Experiments 2A and 2B) and indeed even the participants were identical. Is there an independent justification for selecting different time-windows beyond visual inspection? Would the same results obtain with identical time-windows across the two experiments?

We reanalyzed the data using identical time windows for Experiment 2A and 2B. The result patterns did not change.

Differences between frequent vs. deviant-recombined stimuli were with a more negative response in infants that is late vs. a more positive response in adults. why do the authors conclude that these are equivalent or evidence of a similar learning mechanism?

We base this interpretation on the common observation that what are negative effects in infants or children are often positive effects in adults and vice versa. For example, Kouider et al., (2015) recorded a “surprise” slow negativity in infants which corresponds to a typical P3 (slow positive wave) in adults. Moreover, latencies of experimental effects are known to decrease with increasing age. We link effects in infants and adults based on the eliciting experimental manipulation and relative timing (early vs. late). We added a short paragraph on this issue at the end of the Discussion section. Finally, typical crossmodal binding effects have been observed to start in adults 200 ms after stimulus onset (Bruns and Röder, (2010) and Bonath et al., (2007)). We do not talk about identical mechanisms though, which would indeed be too strong given the extensive differences between infants’ and adults’ brains.

Discussion section:Early discrimination of standard vs. deviant stimuli. Consider in relation to Kouider et al., 2015 where early components were modulated by cross-modal associations.

Kouider et al., (2015) interpreted their early effects as attentional enhancement. They used a cuing paradigm where the auditory cue predicted the visual stimulus. By contrast, such crossmodal predictions prior to stimulation were not possible in the present study.

Reviewer #2:The authors report that 6-month-old infants rapidly learn cross-modal associations implicitly, while in adults there is no evidence of such learning unless the particular associations are explicitly attended to. This is an impressive study with interesting and important results which add to our knowledge of the organisation of human multisensory processing and learning. I have only minor comments, generally related to structure and presentation.The plan of experiments and results is rather hard to follow and keep in mind. We get methods for multiple experiments using slightly different setups, then a lot of different results (which are not interpreted/summarised in an interim way), before discussion. The rationale for the specific design (why we have those particular three conditions) is nowhere spelled out either. It is a clever design, and a good choice by the authors, but the authors should take some time to explain the logic behind their choice (why these conditions) in the Introduction or Materials and methods section. E.g. there is clearly logic behind matching the (low) frequencies of the "rare recombined" and "deviant" stimuli but the point of this manipulation is never explicitly explained or discussed.

We more explicitly explain the rationality of the design in the revised manuscript. (Introduction; Materials and methods section (Experiment 1A and 1B; Experiment 2).

I would suggest a few things to make the paper more accessible.1) Presenting the experiments as a sequence – with 2A and 2B as follow-ups after interim conclusions based on Experiment 1, rather than the (not very plausible to me) suggestion in intro that they were all planned and predicted from the outset, as if Experiments 2A and 2B were run without knowing any results from Experiment 1.

We changed the organization of the manuscript accordingly. The original order actually had historical reasons since we indeed ran the adult experiments 2A and 2B first followed by Experiment 1a and Experiment 1b (the latter to link Experiment 1 to Experiment 2). The main reason why we started with the adult study was to make sure that the paradigm in principle is useful before running the time consuming infant study. However, we think that the reader will easier understand the rationale of the 2*2 experiments in the suggested and now new organization of the manuscript.

2) As suggested above, explaining rationale for the choice of different conditions and how they are matched (e.g. in terms of probability) more clearly when these are introduced.

We added a more detailed description of the design at the end of the Introduction.

3) Within the Results section, add some interim summaries of what each set of results/analyses has shown ("In summary…").

We added interim Results sections (see Results section of the revised manuscript).

4) Consider some kind of table or other way of summarising the results of the different analyses. I found myself writing the 3 condition names (FS / RR / RD) for each analysis and joining these with lines to show where some differed – one easily visualisable outcome from this is that many clusters / epochs / ages show a FS-RD difference whereas the FS-RR difference (of most interest) is much rarer. A high-level table or visualisation that makes this clear would be helpful.

We summarize the main findings of all four experiments (1A/B and 2A/B) in the new Table 2 of the revised manuscript. We refer to this table at the beginning of the Discussion section when we summarize the main results.

My other substantive comment is on power of the adult vs infant analysis to find the effects of interest – this is mentioned in the Discussion section, but the argument would be more convincing with a quantitative treatment (e.g. comparing the size of effect that was detected successfully with adults with the size of effect that was seen in infants but not detected in adults).

We calculated effect sizes for Experiment 1A and 1B. The effect size for the ‘Rare recombined stimuli’ deviant effect of Experiment 1A was larger than the effect size for the rare deviant stimulus effect in both infants (Experiment 1A) and adults (Experiment 1B). Thus, power is unlikely an explanation for the missing ‘Rare recombined stimulus’ effect in Experiment 1b since we were able to detect an effect of a smaller size. We added these calculations to the Results section (Experiment 1).

Reviewer #3:The authors show that infants can learn a cross-modal predictive mapping implicitly while adults can only learn when the mapping is task-relevant, demonstrating superior cross-modal plasticity in infants.This is potentially an interesting finding although the paper does not strike me as being optimally written for eLife. The paper is very long, structured like a classical experimental psychology paper and contains many details, which is not the standard for this kind of papers in eLife. I would strongly recommend shortening it and maybe presenting it as a brief article because at the end the main message is lost in many details which are much less relevant.

We report four extensive EEG experiments including one experiment with infants. Since we directly compared the results across these experiments we presented them in a very standardized manner and in a way that the reader would be easily able to recognize the parallels and crucial differences among the experiments. However, in order to address the needs of those readers of *eLife* who are not interested in all details, we added the basic rationale of the study at the end of the Introduction. Moreover, we included short summaries of the main results both at the beginning and end of each part of the Results section.

Furthermore, as a response to the suggestion of another review we restructured the manuscript and now present first all parts of Experiment 1A/B followed by the main ideas and a presentation of Experiment 2A/B. Finally, we added a new Table 2 which graphically summarizes the main results of all four experiments.

Another major issue is the design and analysis of the data. As I understand it, the baseline correction is made out of the period just preceding the visual stimulus. This would mean that it includes auditory potentials. The baseline should be subtracted in the period before the whole pair of stimuli, otherwise the visual ERPs will necessarily be contaminated by the (inverse of) the auditory potentials. This might fully explain the effect induced by the rare deviant stimuli condition, where there was also a new auditory stimulus (A3).

In the revised manuscript we make more explicit that the auditory and visual elements of a crossmodal stimulus were presented simultaneous, that is, with the same onset too (Materials and methods section). Thus, the baseline used is the epoch preceding both the visual and auditory elements of a crossmodal stimulus.

Another issue is about the region of interest approach. Because the analysis is made on the visual stimuli, one might expect a focus on visual electrodes and their modulation as a function of condition. For infants, the cluster CP seems to show an effect, but it does not appear on the figures of the ERPs.

In order to show the complete topography of the experimental effects we use topographic maps. We report and mark all clusters/electrodes were effects were significant in the text and in the new Table 2, respectively. In addition, we display ERPs only for selected clusters a) to show “real” ERPs what we consider essential for allowing the reader to evaluate the quality of the recordings (morphology of the ERP, timing of the effects etc.); b) in order to avoid overwhelming the reader unfamiliar with ERPs with too many details not essential for the understanding of the results. We report ERPs to synchronously presented crossmodal (AV) stimuli while Emberson et al., (2015) analyzed NIRS-activity to visual vs. omitted visual stimuli; Kouider et al., (2015) looked at ERPs to visual stimuli correctly or incorrectly cued by a preceding sound.